# Inhibition of sterol O-acyltransferase 1 blocks Zika virus infection in cell lines and cerebral organoids
Anja Schöbel[1], Vinicius Pinho dos Reis[1], Rabea Burkhard[1], Julia Hehner[1], Laura Schneider [ORCID][1], Martin Schauflinger[1], Gabrielle Vieyres[2] & Eva Herker [ORCID][1] [✉]

Viruses depend on host metabolic pathways and flaviviruses are specifically linked to lipid metabolism. During dengue virus infection lipid droplets are degraded to fuel replication and Zika virus (ZIKV) infection depends on triglyceride biosynthesis. Here, we systematically investigated the neutral lipid–synthesizing enzymes diacylglycerol O-acyltransferases (DGAT) and the sterol O-acyltransferase (SOAT) 1 in orthoflavivirus infection. Downregulation of DGAT1 and SOAT1 compromises ZIKV infection in hepatoma cells but only SOAT1 and not DGAT inhibitor treatment reduces ZIKV infection. DGAT1 interacts with the ZIKV capsid protein, indicating that protein interaction might be required for ZIKV replication. Importantly, inhibition of SOAT1 severely impairs ZIKV infection in neural cell culture models and cerebral organoids. SOAT1 inhibitor treatment decreases extracellular viral RNA and E protein level and lowers the specific infectivity of virions, indicating that ZIKV morphogenesis is compromised, likely due to accumulation of free cholesterol. Our findings provide insights into the importance of cholesterol and cholesterol ester balance for efficient ZIKV replication and implicate SOAT1 as an antiviral target.

Zika virus (ZIKV), a human pathogenic mosquito-borne orthoflavivirus within the *Flaviviridae* family has recently spread from Africa and Asia to the Pacific and Americas, causing several outbreaks from 2007 onwards[1]. As of 2021, 89 territories and countries have reported mosquito-to-human ZIKV transmission (ZIKA EPIDEMIOLOGY UPDATE, WHO, 2022), and sexual as well as mother-to-fetus transmission have been described[1]. Although ZIKV infections usually pass asymptomatically or with mild flu-like symptoms, recent cases have been associated with new clinical manifestations, including severe neurological complications in adults as well as pregnancy loss and fetal microcephaly[2–4]. Flaviviruses alter lipid metabolic pathways to facilitate efficient viral replication[5–12]. Orthoflavivirus replication organelles are formed by membrane invaginations of the endoplasmic reticulum (ER)[13] that have a distinct lipid profile. Lipid droplets (LDs), the cellular storage organelles of neutral lipids, triglycerides (TGs) and cholesterol esters (CEs), are essential for replication of different pathogens, including the hepatitis C virus (HCV)[14,15], a close relative to ZIKV, and lipophagic degradation of LDs is key to efficient replication of dengue virus (DENV)[9,12]. TG levels increase in ZIKV-infected placenta models[8] and hepatoma cells[11], and the TG-synthesizing diacylglycerol O-acyltransferase 1 (DGAT1) was identified as a host dependency factor for ZIKV replication in placenta cells[8], neural cells, and murine models[10]. Cellular cholesterol is crucial for replication of various viruses[16], including flaviviruses[17]. Within

the *Orthoflavivirus* genus, DENV and West Nile virus (WNV) have been described to accumulate free cholesterol at their replication sites[18,19]. While statin treatment reduced ZIKV, WNV, and DENV infection[19–21], supplementation with cholesterol decreased DENV, Japanese encephalitis virus (JEV), and WNV infection, highlighting a strong dependency of orthoflaviviruses on tightly balanced cholesterol levels[19,22]. To maintain physiological levels of free cholesterol, excess cholesterol is esterified by the sterol O-acyltransferases (SOAT) 1 and 2 and synthesized CEs are stored in LDs[23]. Pharmacological inhibition of SOAT1 and 2 significantly decreased hepatitis B (HBV) and HCV particle production[24,25] and blocked entry as well as RNA replication of SARS-CoV-2[26].

Here, we first used an RNAi-based approach to investigate the role of the neutral lipid–synthesizing enzymes DGAT1 and 2, and SOAT1 in orthoflavivirus replication. In follow-up experiments we specifically identified SOAT1 as a druggable target to limit ZIKV infection in diverse cell lines as well as human cerebral organoids.

## Results

### SOAT1 is a pan-orthoflaviviral host dependency factor

To systematically investigate a potential role of the main neutral lipid–synthesizing enzymes in orthoflavivirus infection, we used RNAi to target the TG-synthesizing enzymes DGAT1 and DGAT2 as well as the

[1]Institute of Virology, Philipps-University Marburg, Marburg, Germany. [2]Institute of Virology and Cell Biology, University of Lübeck, Lübeck, Germany. [✉]e-mail: eva.herker@uni-marburg.de

cholesterol-esterifying SOAT1 (Fig. 1a). We introduced target-specific shRNAs or a non-targeting (shNT) control into Huh7 hepatoma cells via lentiviral transduction and confirmed successful and stable downregulation of either enzyme via quantitative RT PCR (qRT-PCR) and immunoblot analysis (Fig. 1b). Of note, SOAT2-specific shRNAs as well as inhibitors that block SOAT1 and 2 were cytotoxic in Huh7 cells. Due to lack of a functional antibody, knockdown efficiency of DGAT2 was validated only on mRNA

level. Next, we infected Huh7 knockdown cells with DENV-2, tick-borne encephalitis virus Neudoerfl (TBEV), WNV New York 99, or ZIKV PRVABC59 and analyzed viral titers at 48 h post infection (hpi). Depletion of SOAT1 reduced DENV, TBEV, WNV, and ZIKV titers while depletion of DGAT1 and DGAT2 only affected ZIKV (Fig. 1c). Additionally, we analyzed viral envelope (E) protein level in knockdown cells at 48 hpi (Fig. 1d). In line with reduced viral titers, viral E protein levels were slightly lower in

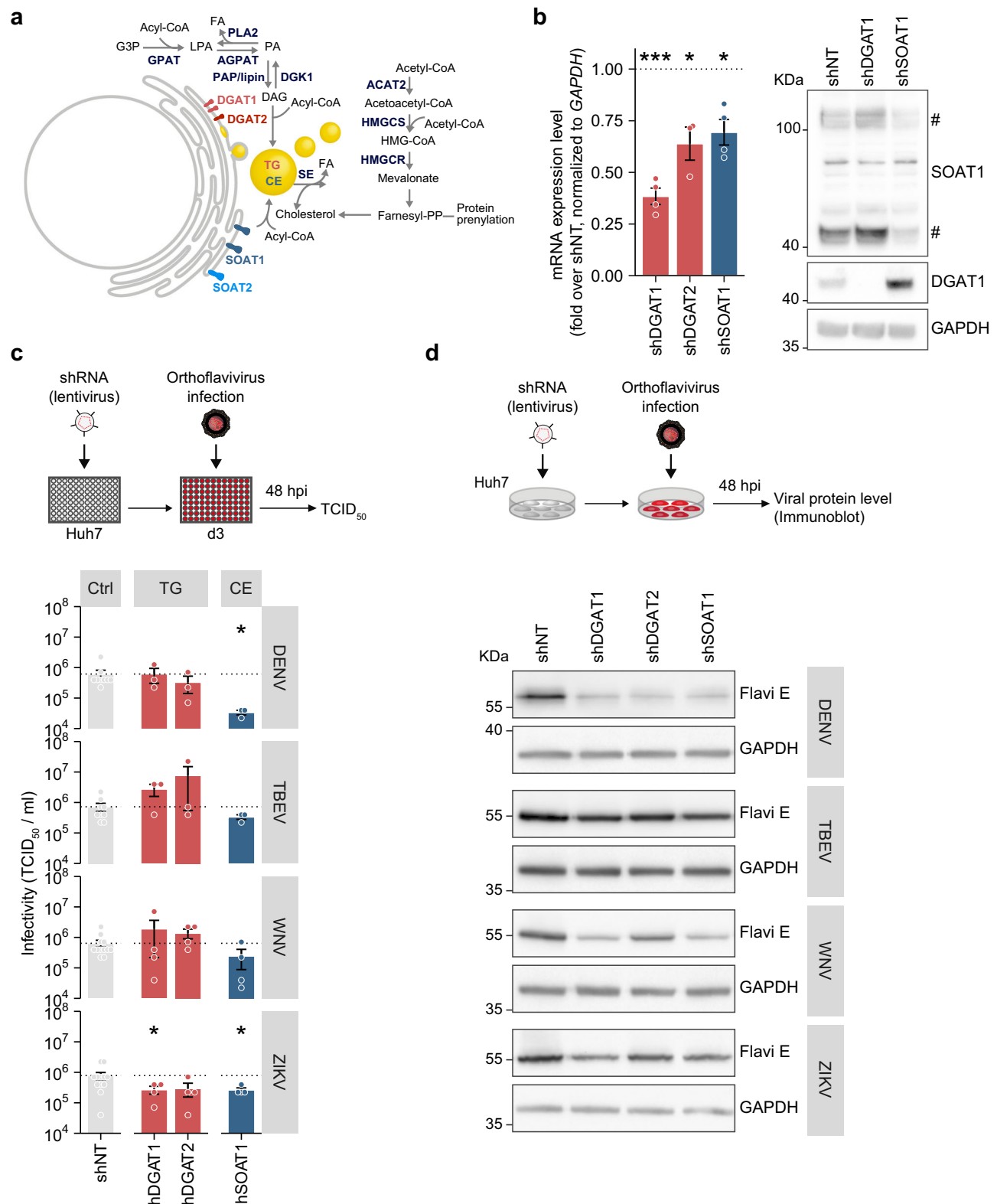

**Fig. 1 | Neutral lipid–synthesizing enzymes are host dependency factors for ZIKV infection. a** Neutral lipid synthesis pathway: the diacylglycerol O-acyltransferases (DGAT) 1 and 2 use diacylglycerol and acyl-CoA as substrate to catalyze triglyceride (TG) formation. The sterol-O-acyltransferases (SOAT) esterify cholesterol. Neutral lipids are stored in cellular LDs. **b** Validation of shRNAs targeting DGAT1, DGAT2, or SOAT1 in ZIKV-infected cells. Huh7 cells were transduced with lentiviral particles encoding shDGAT1, shDGAT2, shSOAT1, or a non-targeting shRNA (shNT). Cells were infected at 3 days post transduction (dpt) with ZIKV (MOI 0.2), and total RNA was isolated at 48 hpi / 5 dpt. mRNA expression levels of the respective target were determined by quantitative RT PCR (qRT-PCR) ($n = 3–4$, $*p ≤ 0.05$, $***p ≤ 0.001$, one-sample $t$-test). Knockdown of DGAT1 and SOAT1 was additionally confirmed by immunoblotting using SOAT1- and DGAT1-specific antibodies. GAPDH served as loading control. # indicates SOAT1-specific bands ($n = 3$). **c** Huh7 cells were transduced with lentiviral particles expressing shRNAs. 3 dpt, cells were infected with DENV (MOI 0.05), TBEV (MOI 0.05), WNV (MOI 0.001), or ZIKV (MOI 0.1). Supernatants were harvested at 48 hpi and titers were determined using TCID$_{50}$ (mean ± SEM, $n = 3–4$, $*p ≤ 0.05$, no asterisk = not significant, Welch´s $t$-test). Note that 3 replicates of shNT were included in each individual experiment. TG triglyceride, CE cholesterol ester. **d** Huh7 knockdown cells were infected with DENV (MOI 0.1), TBEV (MOI 0.1), WNV (MOI 0.001), or ZIKV (MOI 0.1) at 3 or 4 dpt. Lysates were harvested at 48 hpi and viral envelope (Flavi E) protein level were analyzed by immunoblotting. GAPDH served as loading control (Shown is one representative experiment, $n = 2$).

cells depleted of SOAT1. Although mostly not affecting viral titers, the knockdown of DGAT1 also lead to a reduction of E protein level. Taken together these results indicate that SOAT1 might be a pan-orthoflavivirus host factor.

## DGAT1 and SOAT1 are required for efficient ZIKV replication and cytopathic effects

As we observed a concurrent reduction of ZIKV titers and ZIKV E protein in DGAT and SOAT knockdown cells, we decided to further investigate the role of these enzymes in ZIKV replication. Thus, we compared ZIKV intracellular RNA and protein level in shRNA-expressing cells (Fig. 2a). Knockdown of DGAT1, but not DGAT2, strongly decreased ZIKV genome copies, while SOAT1 depletion was less effective (Fig. 2b). Correlating with significantly decreased ZIKV genome copies, we observed reduced levels of the viral E, C, as well as non-structural (NS) 1 protein in shDGAT1- and shSOAT1-, but not shDGAT2-expressing cells (Fig. 2c). As progressive ZIKV infection can cause substantial cytopathic effects (CPE), we next assessed ZIKV-induced CPE. Here, we observed significantly reduced CPE in DGAT1- and SOAT1-depleted cells (Fig. 2d), indicating impaired cell death or viral replication. Although we did observe slightly reduced ZIKV titers in DGAT2-depleted cells, CPE was similar to shNT-expressing cells, again indicating a less important role in ZIKV replication compared to DGAT1 and SOAT1. In line with increased cell survival, we observed a reduced number and substantially smaller ZIKV plaques in shDGAT1- and shSOAT1-transduced cells. In contrast, plaque number and morphology did not drastically change after DGAT2 knockdown (Fig. 2e). Taken together, our data suggest a proviral role for DGAT1 and SOAT1 in ZIKV infection in hepatoma cells.

## Enzymatic activity is required for the proviral function of SOAT1 but not of DGAT1

As our results suggested that SOAT1 and DGAT1 are host-dependency factors for ZIKV replication, we next addressed if their enzymatic activity is required by using commercially available inhibitors for DGAT1 and DGAT2, and two different SOAT1-specific inhibitors (K604 and ATR-101). After confirmation of cell viability (Supplementary Fig. 1a), we analyzed ZIKV infection in inhibitor-treated Huh7 cells (Fig. 3a). Contrary to the pronounced reduction of ZIKV replication in DGAT1 depleted cells, inhibition of DGAT1 activity slightly increased viral protein level in cell lysates (Fig. 3b) and did not change viral titers (Fig. 3c). In addition, inhibition of DGAT2 or combined treatment with both DGAT inhibitors did not reduce ZIKV infection, but marginally increased intracellular viral protein level, despite near-complete loss of LDs, which demonstrates complete inhibition of DGAT activity (Supplementary Fig. 1b).

Treatment with ATR-101 dose-dependently reduced intracellular ZIKV protein level, whereas both SOAT1 inhibitors reduced ZIKV E protein in cell culture supernatants in a dose-dependent manner (Fig. 3b), but viral titers were only marginally reduced in ATR-101-treated cells (Fig. 3c). We additionally performed CPE assays (Fig. 3d). Importantly, inhibitor treatment itself did not reduce cell growth (Supplementary Fig. 1c). Although we did not observe any effect on viral titers, combined inhibition

of both DGAT enzymes slightly reduced ZIKV-induced cell death. In addition, we observed that treatment with high dose of the SOAT1i K604, but not ATR-101, prevented ZIKV-induced CPE. Overall, our inhibitor data indicate that the enzymatic activity of SOAT1 is key to its proviral function, whereas DGAT enzyme activity is not required.

## DGAT1 interacts with the ZIKV capsid protein

Since DGAT1 expression but not its enzyme activity seemed to be critical for efficient ZIKV infection, we hypothesized that a direct interaction of DGAT1 with a viral protein is required for efficient ZIKV replication. We previously have shown that DGAT1 interacts with the HCV capsid protein core[27], a step that is crucial for HCV particle formation. Thus, we investigated a potential interaction between the ZIKV capsid protein C and DGAT1 by co-expressing $^{FLAG}$mDGAT1 and ZIKV C$^{Strep}$ in HEK293T cells and performing co-immunoprecipitation experiments. Indeed, ZIKV C$^{Strep}$ co-purified with $^{FLAG}$mDGAT1, indicating a direct interaction between DGAT1 and ZIKV C protein (Fig. 4a). Concurrently, endogenous DGAT1 co-localized with ZIKV C at LDs in Huh7 cells, presumably at ER-derived membranes surrounding LDs as DGAT1 cannot access the LD surface (Fig. 4b). Thus, it is likely that the interaction between DGAT1 and ZIKV C contributes to efficient ZIKV replication.

## SOAT1 but not DGAT inhibition strongly impairs ZIKV replication in microglia cells

Since ZIKV is neurotropic, we performed experiments in human microglia cells (Hmc3) representing the immune cells of the brain, that have been implicated in ZIKV neuropathology[28]. Whereas liver cells express both SOAT enzymes, human brain cells lack SOAT2 expression[29]. We first analyzed ZIKV protein level and viral titers in Hmc3 after DGAT and SOAT1 inhibition. In line with our observations in hepatoma cells, inhibition of DGAT enzymes did not reduce ZIKV infection (Fig. 5a). Treatment with the SOAT1 inhibitors decreased ZIKV titers, with ATR-101 having the more profound dose-dependent effect. Concomitantly, viral E and C protein level were strongly reduced in cell lysates, and we observed a significant dose-dependent reduction in ZIKV E protein in supernatants, indicating impaired virus production (Fig. 5b). Although to a lesser extent, post-treatment of ZIKV-infected cells with K604 or ATR-101 also reduced viral titers and protein level (Supplementary Fig. 2). As observed in Huh7 cells, DGATi treatment slightly increased ZIKV protein level (Fig. 5b). Both SOAT1 inhibitors dose-dependently decreased ZIKV CPE, likely due to impaired replication (Fig. 5c). Again, single DGAT inhibitor treatment failed to rescue infected cell survival, but in contrast to Huh7 cells, inhibition of both DGAT enzymes did also not reduce ZIKV-induced CPE in Hmc3 cells. Compared to Huh7 cells, the inhibitor effects are more pronounced, indicating that SOAT2 might partially compensate for lack of SOAT1 activity in Huh7 cells.

## SOAT1 inhibition impairs virus production and ZIKV specific infectivity

To further investigate which step in ZIKV replication might be affected by SOAT1 inhibition, we systematically analyzed different steps in the virus

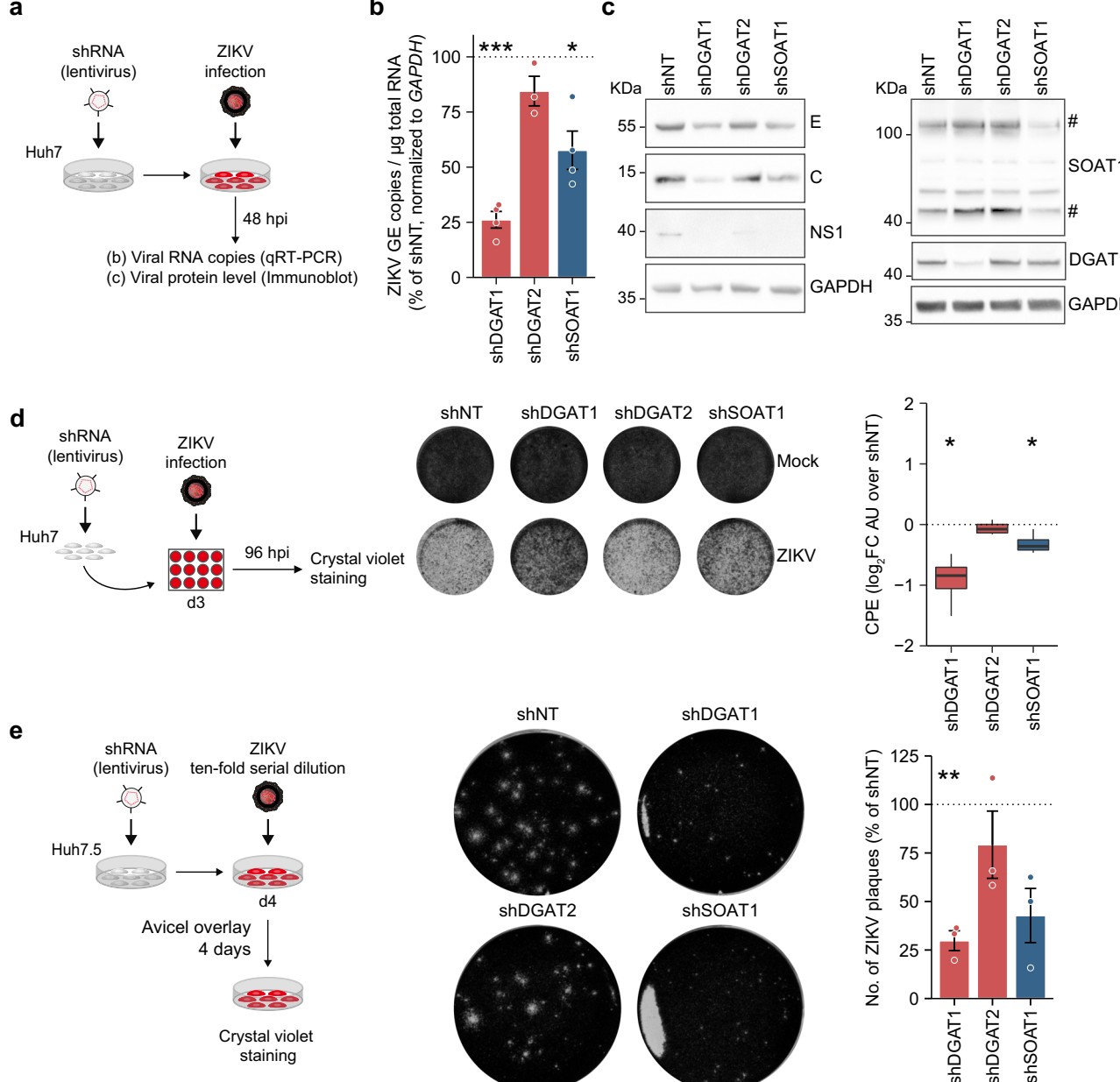

**Fig. 2 | Depletion of DGAT1 and SOAT1 impairs ZIKV replication. a** Scheme of the experimental setup. Huh7 cells were transduced with shRNA lentiviral particles and infected with ZIKV. **b** Cells infected with ZIKV (MOI 0.2) were lysed at 48 hpi and total RNA was isolated. ZIKV copy numbers were quantified by qRT-PCR. Shown is the relative quantification of ZIKV RNA (mean ± SEM, $n = 3$–4, $*p \leq 0.05$, $***p \leq 0.001$, no asterisk = not significant,one-sample $t$-test). **c** shRNA-transduced Huh7 cells were infected with ZIKV (MOI 0.1), lysed at 48 hpi, and analyzed by immunoblotting using a ZIKV C–specific antibody and antibodies against ortho-flaviviral envelope (E) and NS1 proteins. Knockdown of DGAT1 and SOAT1 was confirmed using specific antibodies. # indicates SOAT1-specific bands. GAPDH was used as loading control ($n = 3$). **d** shRNA-expressing Huh7 cells were infected with ZIKV (MOI 0.1) or mock-infected and incubated for 4 d. Cells were fixed and stained with crystal violet to visualize surviving cells. CPE was quantified using Fiji. Box-and-whisker plot indicates CPE as $\log_2$ fold change over shNT (center line: median, box limits: upper and lower quartiles, whiskers: 1.5 x interquartile range, points: outliers, $n = 4$, $*p \leq 0.05$, no asterisk = not significant, one-sample $t$-test). **e** Transduced Huh7.5 cells were infected at 4 dpt with 10-fold serial dilutions of ZIKV virus stocks. The inoculum was removed after 1 h and replaced by Avicel overlay media. Cells were fixed after 4 d and plaque formation was visualized using crystal violet staining. Bar graphs depict the number of ZIKV plaques as percent of shNT (mean ± SEM, $n = 3$, $**p \leq 0.01$, no asterisk = not significant, one-sample $t$-test).

replication cycle (Fig. 6a). First, we addressed a potential effect on virus entry in SOAT1i-treated Hmc3 cells. Here, we did not observe any changes in virus entry compared to DMSO-treated cells (Fig. 6b). We next performed confocal microscopy analysis to visualize the formation of dsRNA foci in Hmc3 cells. For an unbiased approach, we used cells stably expressing a cell-based ZIKV EGFP reporter, as previously described for ZIKV[30] and DENV[31], to select ZIKV-positive cells. In uninfected cells, EGFP is present at the ER but translocates to the nucleus after cleavage by the ZIKV protease.

Neither the quantity nor the size of dsRNA foci were altered in SOAT1i-treated cells compared to DMSO (Fig. 6c), indicating that SOAT1 activity is not required for RNA replication, translation or replication vesicle formation. To confirm our data, we performed transmission electron microscopy (TEM) of ZIKV-infected Hmc3 cells. 24 h post infection, we detected vesicle packets (VP) and virions (Vi) of similar morphology in both DMSO- and ATR-101-treated cells, indicating that replication compartments are still intact after SOAT1 inhibition (Fig. 6d). Virus-induced structures were

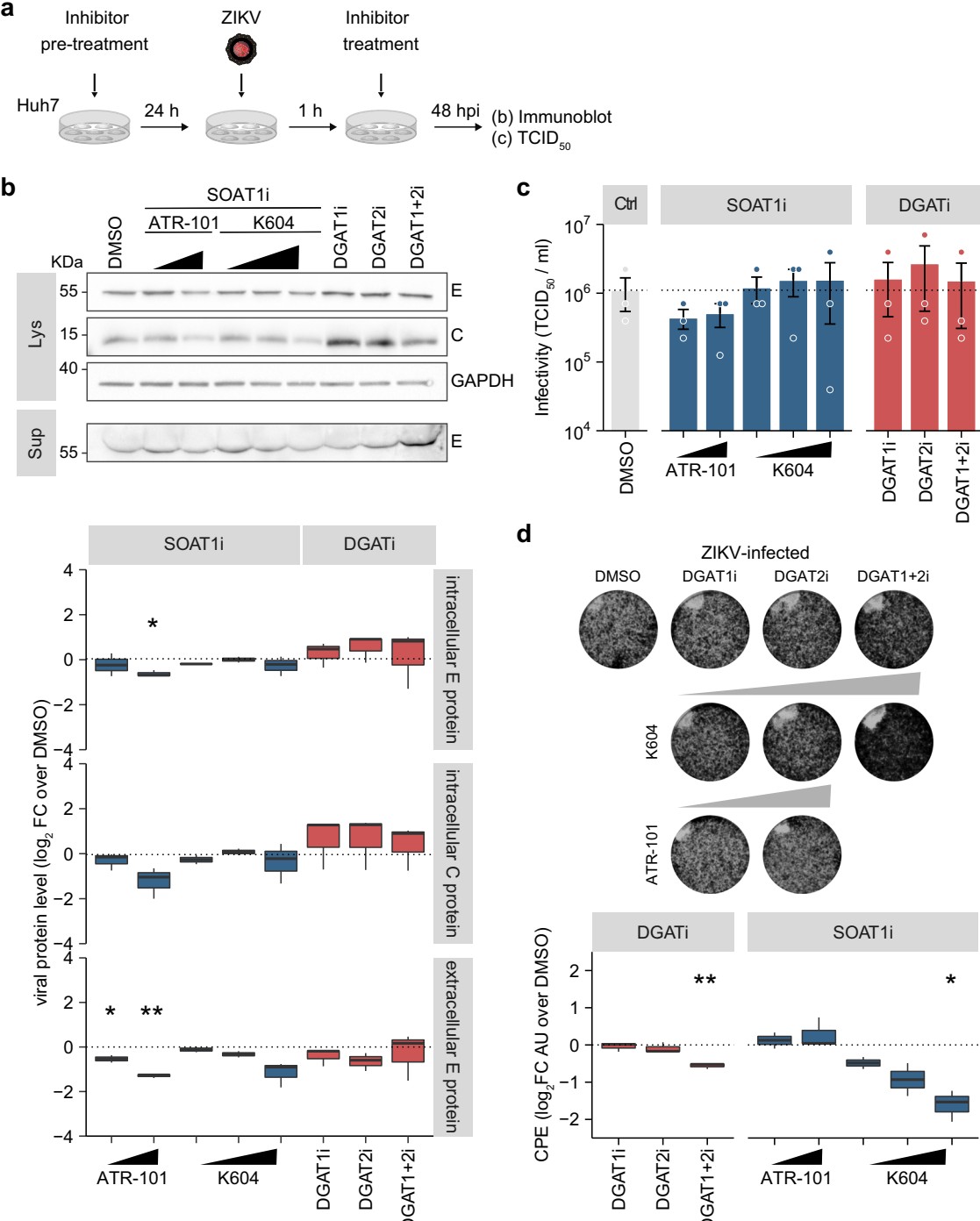

**Fig. 3 | SOAT1 inhibitor treatment, but not DGAT inhibitors reduce ZIKV infection in Huh7 cells. a** Scheme of the experimental setup. Huh7 cells were pre-treated with DGAT inhibitors (5 μM each), SOAT1-specific inhibitors (ATR-101 = 6 μM or 9.72 μM, K604 = 5 μM, 10 μM, or 20 μM), or DMSO 24 h prior to infection with ZIKV (MOI 0.1). After 1 h, the inoculum was removed and replaced by fresh media supplemented with the respective inhibitors. At 48 hpi, supernatants were harvested and cells were lysed. **b** Cell lysates and supernatants were analyzed by immunoblotting using ZIKV C and orthoflavivirus E-protein specific antibodies (Lys = cell lysates, Sup = supernatant). GAPDH served as loading control. Shown is one representative immunoblot. Bands were quantified with ImageLab and intra-cellular signals were normalized to GAPDH. Box-and-whisker plot shows viral

protein level as $\log_2$ fold change over DMSO control (center line: median, box limits: upper and lower quartiles, whiskers: 1.5 x interquartile range, points: outliers, $n = 3$, except for K604 5 μM and 10 μM ($n = 2$), *$p \le 0.05$, **$p \le 0.01$, no asterisk = not significant, one-sample $t$-test). **c** Supernatants from cells in (**b**) were analyzed using $TCID_{50}$ titration (mean ± SEM, $n = 3$, no asterisk = not significant, Welch´s $t$-test). **d** After 4 d, cells were fixed, surviving cells were visualized by crystal violet staining, and CPE was quantified using Fiji. Box-and-whisker plot indicates CPE as $\log_2$ fold change over DMSO control (center line: median, box limits: upper and lower quartiles, whiskers: 1.5 x interquartile range, points: outliers, $n = 3$, except for K604 5 μM and 10 μM ($n = 2$), *$p \le 0.05$, **$p \le 0.01$, no asterisk = not significant, one-sample t-test).

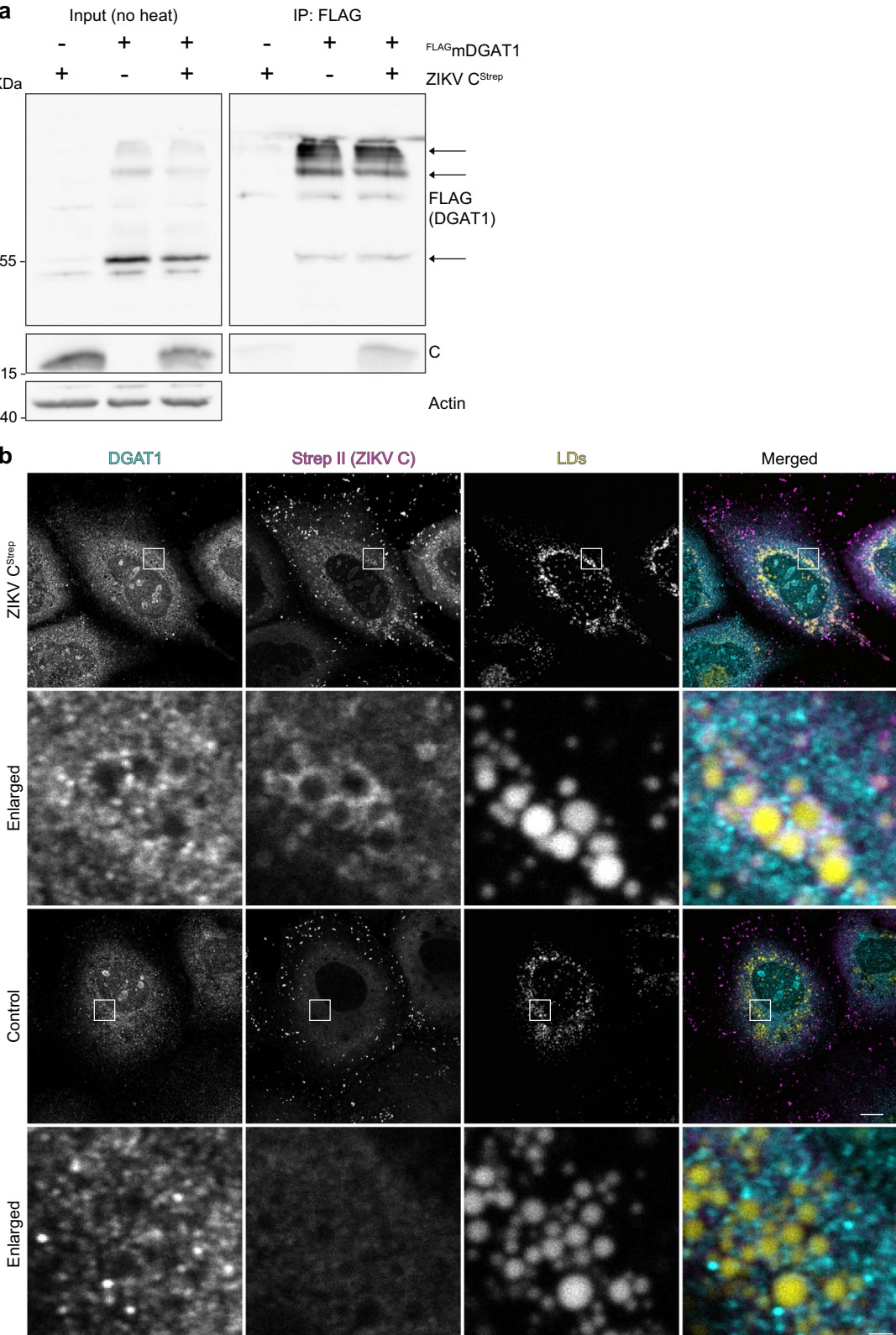

**Fig. 4 | The ZIKV capsid protein interacts with DGAT1. a** Co-immunoprecipitation of $^{FLAG}$mDGAT1 and ZIKV MR766 C$^{Strep}$ (-anchor). HEK293T cells were transiently co-transfected with $^{FLAG}$mDGAT1 and ZIKV MR766 C$^{Strep}$-expressing plasmids. Cells were lysed 2 days post transfection and clarified lysates were subjected to FLAG-specific immunoprecipitation. Lysates (Input) and immunoprecipitated (IP) samples were analyzed by immunoblotting. Note that input samples were not heated to prevent aggregation of DGAT1. Arrows indicate DGAT1-specific bands. Actin served as loading control. Shown is one representative immunoblot ($n = 2$). **b** DGAT1 co-localizes with ZIKV C at LD-surrounding membranes. Huh7 cells were transfected with a ZIKV PRVABC59 C$^{Strep}$ ( + anchor)-expression plasmid, fed with oleic acid the following day, and fixed after 2 days. C and DGAT1 were stained using DGAT1- and Strep II-specific antibodies. Empty vector-transfected cells served as control. LDs were visualized using BODIPY 655/676. Shown are representative images ($n = 2$; scale bar 10 μm, scale bar inset 1 μm).

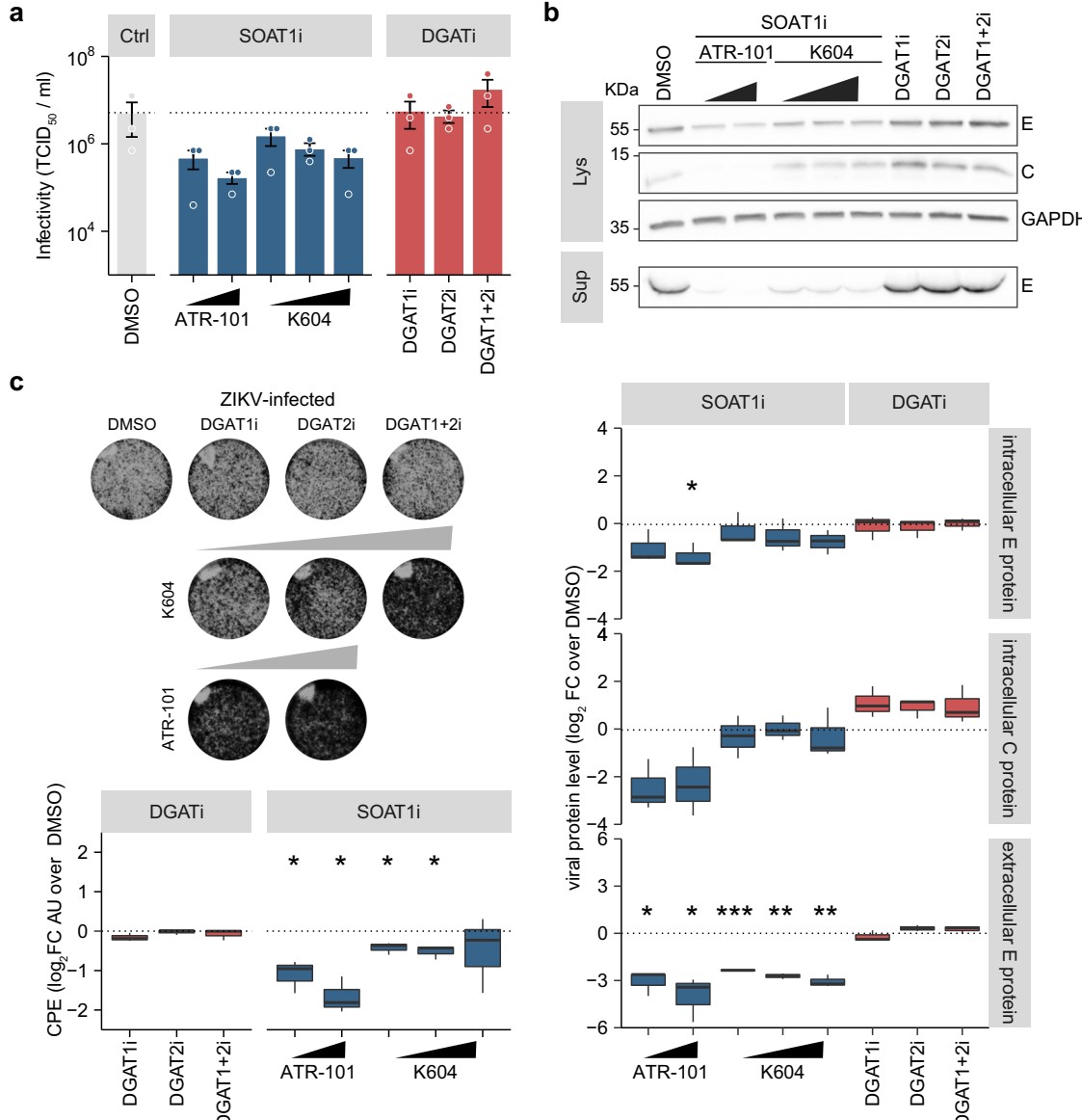

**Fig. 5 | Inhibition of SOAT1 reduces ZIKV infection in human microglia cells.**
**a**, **b** Hmc3 cells were pre-treated with DGAT inhibitors (each 5 μM), SOAT1-specific inhibitors (ATR-101 = 6 or 9.72 μM, K604 = 5, 10, or 20 μM) or DMSO 24 h prior to infection with ZIKV (MOI 0.0025). After 1 h, the inoculum was removed and replaced by fresh media supplemented with the respective inhibitors. Cell lysates and supernatants were harvested at 48 hpi. Viral titers were analyzed using $TCID_{50}$ (mean ± SEM, $n = 3$, no asterisk = not significant, Welch´s $t$-test) (**a**). Corresponding viral protein levels were analyzed by immunoblot using ZIKV C and orthoflaviviral E protein-specific antibodies. GAPDH served as loading control (**b**) (Lys = cell lysates, Sup = supernatant). Bands were quantified with ImageLab and intracellular signals were normalized to GAPDH. Box-and-whisker plot shows viral protein level as $\log_2$ fold change over DMSO control (center line: median, box limits: upper and lower quartiles, whiskers: 1.5 x interquartile range, points: outliers, $n = 3$, *$p \leq 0.05$, **$p \leq 0.01$, ***$p \leq 0.001$, no asterisk = not significant, one-sample $t$-test).
**c** Treatment with SOAT1i reduces ZIKV-induced CPE in Hmc3 cells. Cells were treated as described above, followed by infection with ZIKV (MOI 0.01) and re-treatment as described. 96 hpi, cells were fixed and CPE was visualized by crystal violet staining. CPE quantification was performed with Fiji. Box-and-whisker plot indicates CPE as $\log_2$ fold change over DMSO control (center line: median, box limits: upper and lower quartiles, whiskers: 1.5 x interquartile range, points: outliers, $n = 3$, *$p \leq 0.05$, no asterisk = not significant, one-sample $t$-test).

clearly distinguishable from cellular compartments as shown by TEM images of ZIKV-infected and uninfected Hmc3 cells (Supplementary Fig. 3). In the immunoblot analysis, we detected significantly less E protein in supernatants of SOAT1i-treated Hmc3 cells (Fig. 5b), indicating that virus assembly or release might be impaired. In line, we also detected significantly less viral RNA in supernatants of SOAT1i-treated cells (Fig. 6e). Moreover, lack of SOAT1 activity led to a significant reduction in specific infectivity of virions (Fig. 6f). Taken together, our data indicate that inhibition of SOAT1 impairs late steps in the ZIKV replication cycle and the secreted virions are less infectious. However, if assembly or egress are affected remains to be elucidated.

## SOAT1 activity is required for ZIKV replication in astrocytoma cells

To investigate the SOAT1-dependency of ZIKV in other neural cell types, we next infected astrocytoma (1321N1) and neuroblastoma cells (SH-SY5Y) and treated them with DGAT and SOAT1 inhibitors. In line with our observations in Hmc3 cells, SOAT1i-treatment dose-dependently reduced ZIKV titers and ATR-101 treatment decreased intra- and extracellular viral protein level in 1321N1 cells, whereas inhibition of DGAT did not change intracellular proteins, and increased extracellular E protein level in case of DGAT2i and combined DGATi treatment (Fig. 7a, b). In SH-SY5Y cells, SOAT1 inhibition with high dose ATR-101

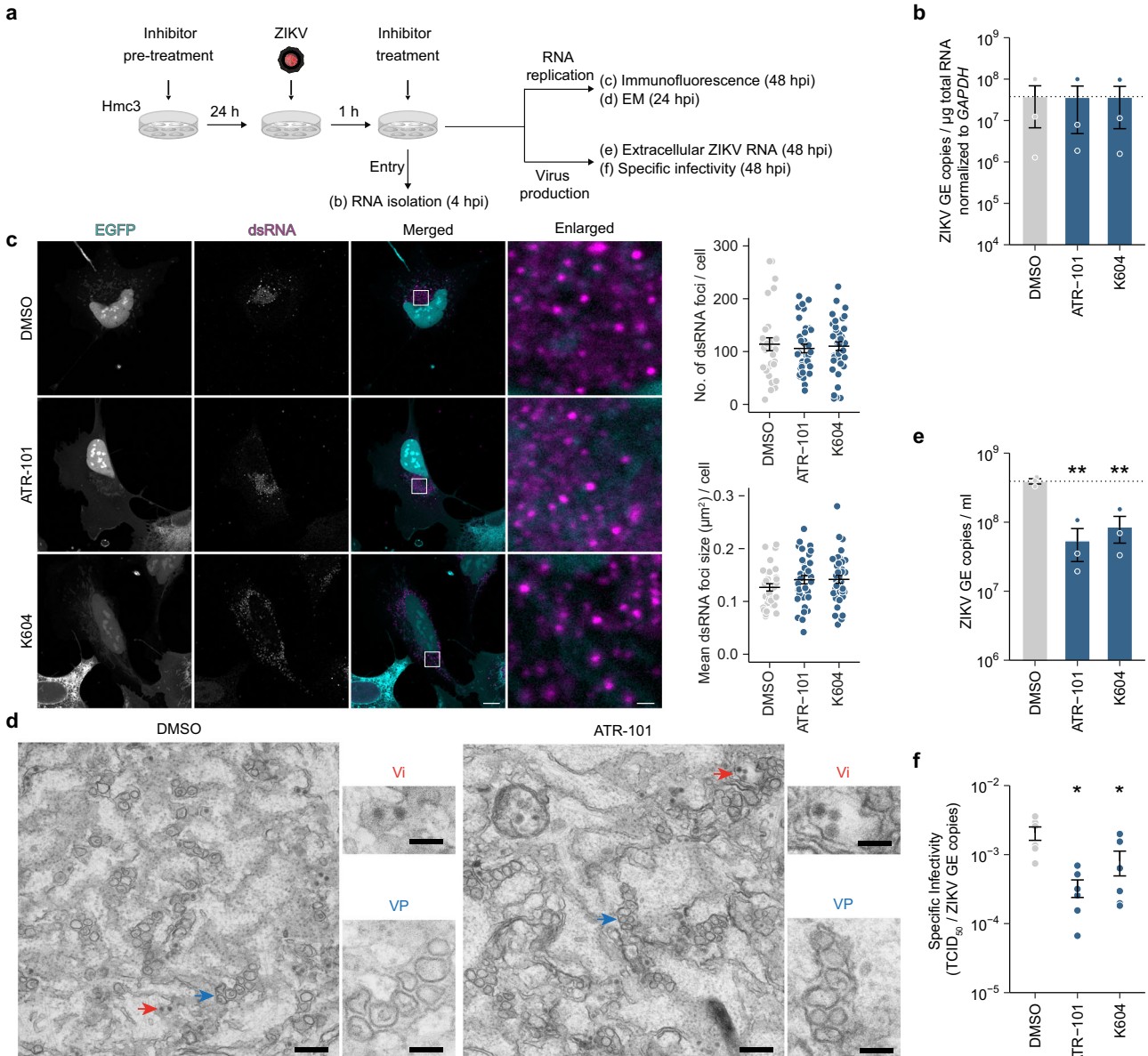

**Fig. 6 | Lack of SOAT1 activity impairs virion production and ZIKV specific infectivity. a** Scheme of experimental setup to decipher the dependency of different ZIKV replication steps on SOAT1. Hmc3 cells were pre-treated with SOAT1-specific inhibitors (9.72 μM ATR-101 or 20 μM K604) or DMSO 24 h prior to infection with ZIKV. After 1 h, the inoculum was removed and replaced by fresh media supplemented with the respective inhibitors and different steps of virus replication were analyzed. **b** To determine viral entry, cells were infected with ZIKV (MOI 5) 24 h after initial treatment. 4 hpi, total RNA was isolated and ZIKV GE copies were determined by qRT-PCR. Shown is the relative quantification of ZIKV RNA (mean ± SEM, *n* = 3, no asterisk = not significant, Welch´s *t*-test). **c** Hmc3 ZIKV EGFP reporter cells were pre-treated with 9.72 μM ATR-101, 20 μM K604, or DMSO for 24 h, infected with ZIKV (MOI 0.0025) and fixed 48 hpi. Cells were stained using a dsRNA-specific antibody and analyzed by confocal microscopy. ZIKV-infected cells were randomly selected by the nuclear translocation of the EGFP reporter.

Shown are representative images (scale bar 10 μm, scale bar inset 1 μm). dsRNA signals from two independent experiments were quantified using the particle analyzer function of Fiji (# of cells $n_{DMSO}$ = 31, $n_{ATR-101}$ = 35, $n_{K604}$ = 39, mean ± SEM, no asterisk = not significant, Welch´s *t*-test). **d** Hmc3 reporter cells were pre-treated with 9.72 μM ATR-101 or DMSO, infected with ZIKV (MOI = 0.1), fixed at 24 hpi, and processed for EM. Shown are representative images (scale bars = 250 nm). Magnifications show vesicle packets (VP) and virions (Vi) (scale bars = 100 nm). **e** ZIKV RNA copies in supernatants of SOAT1i-treated cells were determined by qRT-PCR at 48 hpi (ZIKV MOI 0.0025). Shown are ZIKV GE copies/ml supernatant (mean ± SEM, *n* = 3, **$p ≤ 0.01$, Welch´s *t*-test). **f** Calculation of ZIKV specific infectivity (ratio of $TCID_{50}$ to ZIKV RNA copy number) of SOAT1i- and DMSO-treated cells at 48 hpi from duplicates of 3 independent experiments (mean ± SEM, *n* = 6, *$p ≤ 0.05$, Welch´s *t*-test).

slightly reduced ZIKV infection, but to a lesser extent compared to the other cell lines we used (Fig. 7 c, d). Interestingly, K604 caused a marginal increase in intracellular viral proteins. These diverging findings might result from different metabolization rates of both inhibitors in SH-SY5Y cells. While inhibition of DGAT1 and DGAT2 marginally affected virus titers, inhibition of DGAT2 increased intracellular ZIKV proteins (Fig. 7c, d). Taken together, whereas DGAT inhibition is not antiviral in

our cell culture models, SOAT1 activity is critical for ZIKV replication in primary target cells of ZIKV infection.

## Cholesterol ester treatment does not rescue SOAT1 inhibitor effects on ZIKV infection

To analyze if cholesterol ester formed by SOAT1 are required for ZIKV replication, we performed rescue assays with cholesterol ester treatment

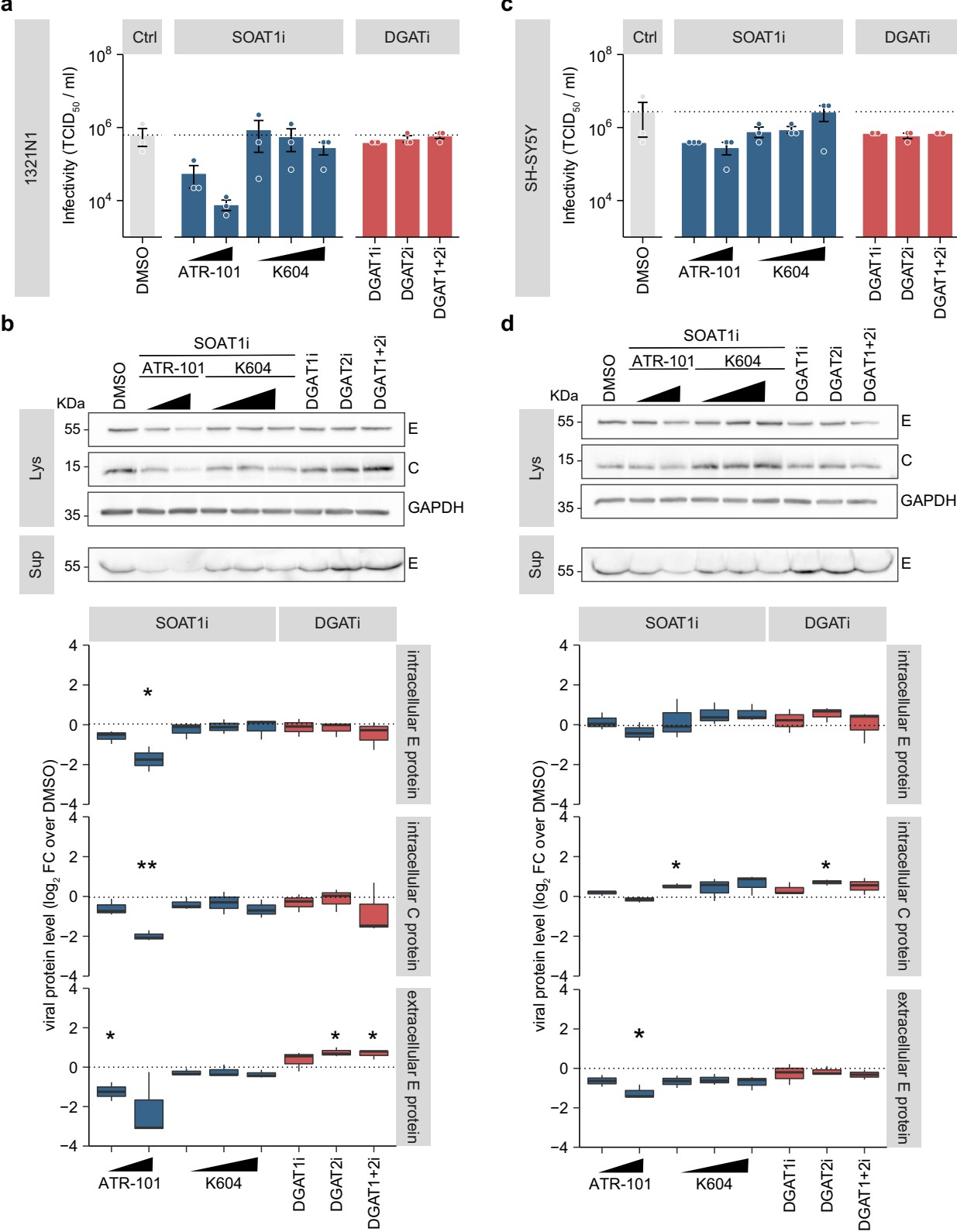

**Fig. 7 | Effect of SOAT1 inhibitor and DGAT inhibitor treatment on ZIKV infection in neural cell lines.** 1321N1 cells (**a**, **b**) and SH-SY5Y cells (**c**, **d**) were pre-treated with DGAT inhibitors (each 5 μM), SOAT1-specific inhibitors (ATR-101 = 6 or 9.72 μM, K604 = 5, 10, or 20 μM) or DMSO 24 h prior to infection with ZIKV ($MOI_{1321N1}$ 0.0125, $MOI_{SH-SY5Y}$ 0.25). After 1 h, the inoculum was replaced by fresh media supplemented with the respective inhibitors. Cell lysates and supernatants were harvested at 48 hpi. Viral titers were determined by $TCID_{50}$ (mean ± SEM, $n = 3$, no asterisk = not significant, Welch´s $t$-test) (**a**, **c**). Corresponding viral protein level were analyzed using ZIKV C and orthoflaviviral E protein-specific antibodies. GAPDH served as loading control (Lys = cell lysates, Sup = supernatant) (**b**, **d**). Bands were quantified with ImageLab and intracellular signals were normalized to GAPDH. Box-and-whisker plot shows viral protein level as $log_2$ fold change over DMSO control (center line: median, box limits: upper and lower quartiles, whiskers: 1.5 x interquartile range, points: outliers, $n = 3$, *$p \leq 0.05$, **$p \leq 0.01$, no asterisk = not significant, one-sample $t$-test).

using ZIKV CPE as a readout. After validation of cholesterol ester uptake in Hmc3 cells (Supplementary Fig. 4), cells were pre-treated with SOAT1i in media supplemented with cholesteryl oleate or respective solvents. After infection, we treated the cells again, and analyzed ZIKV-induced CPE (Fig. 8a). Supplementation with cholesteryl oleate did not affect ZIKV CPE in SOAT1i-treated cells, indicating that the observed reduction in ZIKV infection after SOAT1 inhibition was not due to reduced cellular levels of cholesterol ester.

### Excess free cholesterol impairs ZIKV infection in Hmc3 and Huh7 cells

Tightly regulated cholesterol levels are essential for successful replication of various viruses, including orthoflaviviruses[17] and treatment with cholesterol suppressed DENV and JEV infection[22]. To investigate if cholesterol imbalance caused by SOAT1 inhibition suppressed ZIKV replication, we first visualized free cellular cholesterol using filipin III and LDs with LD540 (Fig. 8b). Filipin III signals were significantly increased irrespective of the SOAT1 inhibitor used (Fig. 8b). Interestingly, K604-treated Hmc3 cells displayed a dot-like staining of filipin III suggesting accumulation of cholesterol in subcellular structures, whereas ATR-101 did not visibly alter cholesterol localization within the cell. Regarding cholesterol level in cell lysates, only ATR-101 treatment caused a slight increase of free cholesterol compared to the DMSO control and K604 (Fig. 8c). To address whether excess cholesterol affects ZIKV replication, we performed CPE assays and either added cholesterol to the virus inoculum (during entry) or post-infection, after removal of the virus. Whereas CPE was evident in the control, supplementation with cholesterol strongly reduced ZIKV CPE. When cholesterol was added during virus inoculation only small plaques were forming, while post-infection treatment blocked ZIKV replication completely (Fig. 8d). This is in line with our observation that SOAT1 inhibition does not affect viral entry (Fig. 6b).

We next verified these results in Huh7 cells. In line with the results in Hmc3 cells, cholesteryl ester treatment did not affect CPE in SOAT1i-treated Huh7 cells (Supplementary Fig. 5a). Again, quantification of the free cholesterol using filipin III staining revealed a significant increase in free cholesterol in SOAT1i-treated cells compared to the DMSO control (Supplementary Fig. 5b), but only ATR-101 treatment slightly increased free cholesterol measured biochemically in cell lysates (Supplementary Fig. 5c). We observed a ring-like staining of filipin III at the rim of LDs upon ATR-101 treatment, a phenotype that has been reported before for SOATi-treated cells[32]. Similar to Hmc3 cells, treatment with K604 caused accumulation of free cholesterol in dot-like structures (Supplementary Fig. 5b). These findings indicate a different mode of action of ATR-101 and K604. As observed in Hmc3 cells, addition of cholesterol during ZIKV entry reduced CPE in Huh7 cells, with only small plaques visible. But in contrast to Hmc3 cells, post-infection treatment even slightly increased ZIKV CPE (Supplementary Fig. 5d), which may be due to rapid esterification in cells expressing both SOAT1 and 2. Although to a lesser extent compared to acute SOAT1 inhibition, SOAT1-knockdown cells also exhibited elevated level of free cholesterol (Supplementary Fig. 6). Taken together, inhibition of SOAT1 leads to accumulation of free cholesterol. This imbalance between free and esterified cholesterol might result in inhibition of ZIKV infection in SOAT1i-treated cells.

### ZIKV replication in hiPSC-derived cerebral organoids relies on SOAT1 and DGAT activity

Stem cell-derived brain organoids have recently been used as 3D cell culture models to study ZIKV infection in vitro[33–38]. To investigate how pharmacological inhibition of the neutral lipid–synthesizing enzymes affects ZIKV infection in brain organoid models, we generated cerebral organoids from two different human induced pluripotent stem cell (hiPSC) lines (HMGU#1 and TISSUi006A) (Fig. 9a). After differentiation, we assessed gene expression of neural identity markers by qRT-PCR (Supplementary Fig. 7). For the analysis of ZIKV infection, we used cerebral organoids that were differentiated for 30–35 days. 24 h after initial inhibitor-treatment, we infected

cerebral organoids with ZIKV and analyzed viral RNA and protein level at 3 dpi (Fig. 9b). Inhibition of SOAT1 decreased ZIKV RNA level compared to the DMSO control (Fig. 9c). Unexpectedly, we also observed a reduction of ZIKV RNA in organoids treated with a combination of both DGAT inhibitors or with the DGAT2 inhibitor alone. Parallel inhibition of both DGAT enzymes also significantly reduced ZIKV protein level, suggesting that in more complex cell systems and culture conditions TG biosynthesis is required for ZIKV infection and replication (Fig. 9d, e). In line with our previous results, SOAT1i-treated organoids showed strongly decreased E and C protein level compared to the DMSO control (Fig. 9d, e). As observed in the neural cell lines, ATR-101 displayed a stronger effect on ZIKV infection than K604, likely due to different potency or mode of action of the inhibitors.

Adding up to our previous findings, the data suggest SOAT1 as important host factor for ZIKV replication in more complex infection models and that targeting SOAT1 is an interesting approach for anti-flaviviral therapy.

## Discussion

Flaviviruses such as ZIKV, DENV, or the related HCV, remodel cellular lipid metabolism to favor efficient replication but differ in their requirement for specific lipids. Neutral lipid–synthesizing enzymes have been described as host factors for several positive-strand RNA viruses, including ZIKV[8,10], HCV[25,27,39], and SARS-CoV-2[26,40,41], as well as HBV[24]. In this study, we identified SOAT1, and to a lesser extent DGAT1 and DGAT2, as ZIKV host dependency factors, which are especially important in microglia cells and cerebral organoids.

Concurrent with reduced virus production, knockdown of DGAT1 or SOAT1 in Huh7 cells diminished viral RNA and protein level, indicating that loss of either enzyme reduced ZIKV infection rates. The decreased number of ZIKV plaques in DGAT1- and SOAT1-depleted cells suggest defective viral entry, impaired viral replication, or spread. In accordance with the exclusive role of DGAT1 in HCV infection[27], depletion of DGAT2 only marginally reduced ZIKV infection compared to DGAT1, indicating a specific function for DGAT1.

The SOAT1-dependency contrasts with recent results indicating increased ZIKV infection in Huh7.5 SOAT1/2 knockout cells[42]. However, the respective study generated only one monoclonal SOAT1/2 knockout cell line using CRISPR/Cas and it is even stated that the increase in ZIKV infection is likely not due to free cholesterol levels. As Huh7-derived cells are genomically unstable, we generated polyclonal knockdown cells with acute downregulation of the enzymes that we investigate to limit adaptation processes and in addition used two different SOAT1-specific inhibitors. Previous investigation of ZIKV replication in SOAT1-depleted Hela-R19 cells also indicated no dependency[43], but these cells express both SOAT enzymes and it is unclear which one is dominant in esterifying free cholesterol.

In our study, shRNA-mediated downregulation, but not pharmacological inhibition of DGAT1, strongly reduced ZIKV infection in Huh7 cells, and DGAT1 inhibitor treatment did not prevent ZIKV infection in neural cell lines (Hmc3, 1321N1, and SH-SY5Y) or cerebral organoids, indicating a mechanism independent of DGAT1 activity. Instead, we rather observed a trend towards increased ZIKV protein level in DGATi-treated cell cultures. In primary placental cells, inhibition of DGAT1 clearly reduced ZIKV replication[8], suggesting a cell type-specific dependency on DGAT1 activity. Inhibition of DGAT1 has also been shown to decrease ZIKV infection in SH-SY5Y cells[10]; in those cells we observed a slight non-significant decrease in viral titers following DGAT inhibitor treatment. The use of a different inhibitor as well as the high concentrations that were required to restrict ZIKV replication in SH-SY5Y cells might explain the differences in efficacy. In SH-SY5Y cells lacking DGAT2 activity the decreased titers correlated with increased intracellular ZIKV protein levels, indicating that virus egress might be affected. However, the mechanistic details remain to be elucidated. Furthermore, we report a direct interaction between DGAT1 and the ZIKV C protein, a phenotype that has been described before for the HCV

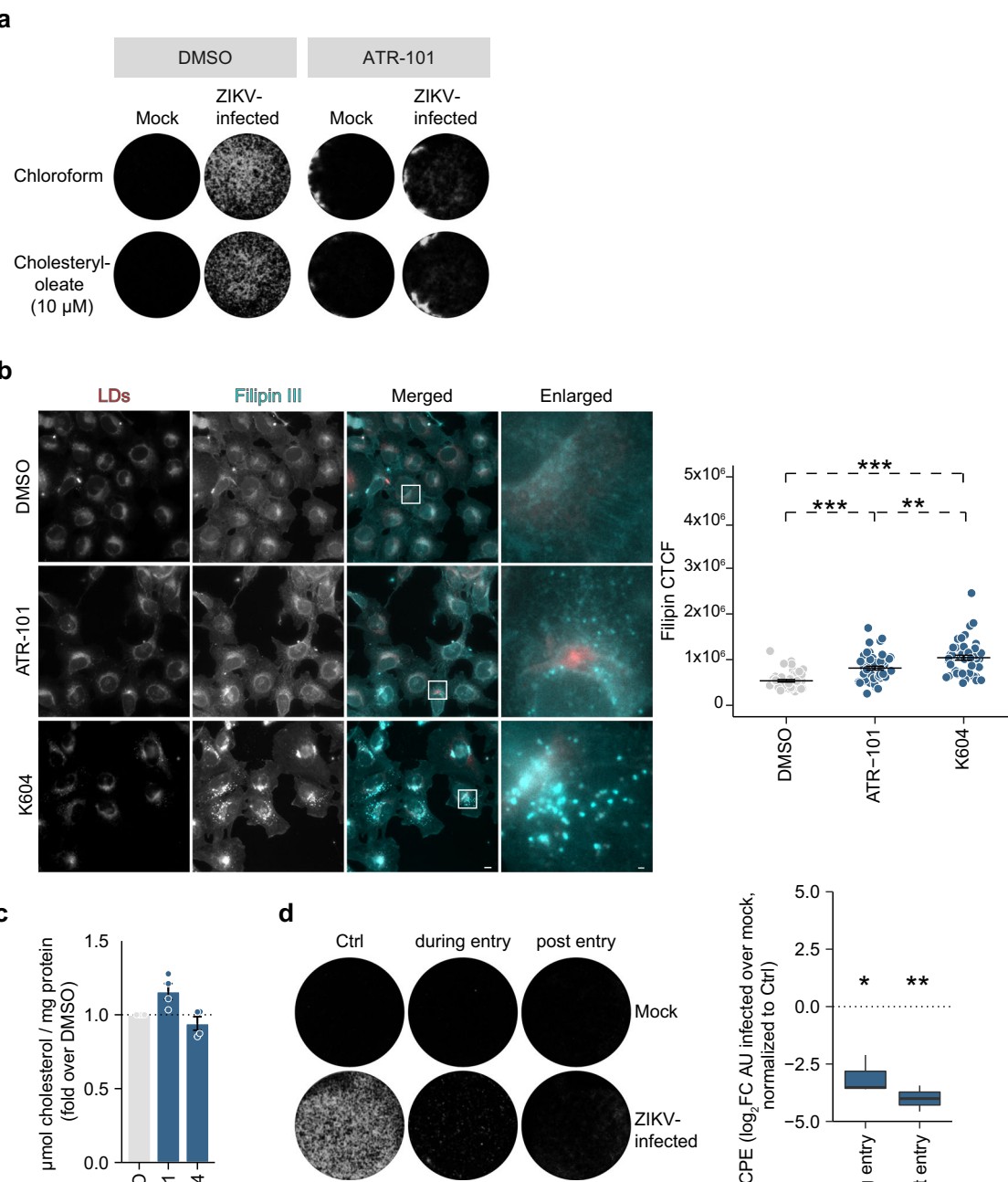

**Fig. 8 | Inhibition of SOAT1 causes free cholesterol accumulation in Hmc3 cells and diminishes ZIKV infection. a** Cells were cultured in serum-free OptiMEM containing DMSO, 20 μM K604, or 9.72 μM ATR-101 and 10 μM cholesteryl oleate or chloroform as vehicle control for 1–2 h before medium was changed to normal growth medium supplemented with the respective inhibitors for additional 22 h. After ZIKV infection (MOI 0.01), cells were re-treated with cholesteryl-oleate or chloroform, before medium was changed to normal growth medium with inhibitors for 3 d. Cells were fixed and CPE was visualized with crystal violet ($n = 2$).
**b, c** Measurement of free cholesterol. Cells were treated with either SOAT1i (9.72 μM ATR-101, 20 μM K604) or DMSO, and fixed after 48 h. LD540 was used to visualize LDs and free cholesterol was stained with filipin III (**b**). Shown are representative images (scale bar 10 μm, scale bar inset 1 μm). The CTCF for the filipin III channel was calculated for individual cells (# of cells $n_{DMSO} = 46$, $n_{ATR-101} = 48$, $n_{K604} = 47$

from 2 independent experiments, mean ± SEM, **$p \leq 0.01$, ***$p \leq 0.001$, Welch's t-test). Cells were treated as described above, lysed, and free cholesterol was measured. Depicted is the relative quantification of free cholesterol normalized to protein content of 2 biological replicates from 2 independent experiments (mean ± SEM) (**c**). **d** Cells were infected with ZIKV (MOI 0.01) or mock-infected and treated with 50 μg/ml water-soluble cholesterol (~ 40 mg of cholesterol per gram; balance methyl-β-cyclodextrin) during virus inoculation (during entry) or after infection (post entry). Cells were fixed after 3 or 4 d, and crystal violet was used to visualize surviving cells. CPE was quantified using Fiji. Box-and-whisker plot indicates CPE as $\log_2$ fold change over mock, normalized to the control (center line: median, box limits: upper and lower quartiles, whiskers: 1.5 x interquartile range, points: outliers, $n = 3$, *$p \leq 0.05$, **$p \leq 0.01$, one-sample t-test).

**Fig. 9 | Inhibition of SOAT1 and DGAT2 reduces ZIKV infection of human iPSC-derived cerebral organoids. a** Differentiation of hIPSCs into cerebral organoids. Two different hIPS cell lines (HMGU#1 and TISSUi006A) were used for differentiation with the StemDiff Cerebral organoid kit. **b** Scheme of the experimental setup. After maturation (day ~35), organoids were pre-treated with DGAT inhibitors (each 5 µM), SOAT1-specific inhibitors (ATR-101 = 9.72 µM, K604 = 20 µM), or DMSO for 24 h, followed by inoculation with $10^5$ TCID$_{50}$ per organoid of ZIKV for 24 h. After removal of the inoculum, organoids were cultured in media supplemented with inhibitors for 3 days. Individual organoids were harvested and infection was analyzed by qRT-PCR and immunoblot. **c** Organoids were lysed and total RNA was isolated at 3 dpi. ZIKV genome copies were analyzed by qRT-PCR. Each dot represents an individual organoid from 3 independent experiments using 2 different cell lines (mean ± SEM, $n = 3$, $*p \le 0.05$, no asterisk = not significant, Welch´s $t$-test). **d** Individual organoids from two independent experiments were lysed at 3 dpi and ZIKV E and C protein level were analyzed by immunoblot. GAPDH served as loading control. Mock-infected and pan-orthoflavivirus inhibitor (NITD008)-treated organoids were used as control. Shown is one representative immunoblot. **e** Band signal intensities were quantified using ImageLab. Box-and-whisker-plot shows ZIKV E and C protein levels as log$_2$ fold change over DMSO normalized to GAPDH (center line: median, box limits: upper and lower quartiles, whiskers: 1.5 x interquartile range, points: outliers, $n = 4$, $*p \le 0.05$, $**p \le 0.01$, no asterisk = not significant, one-sample $t$-test).

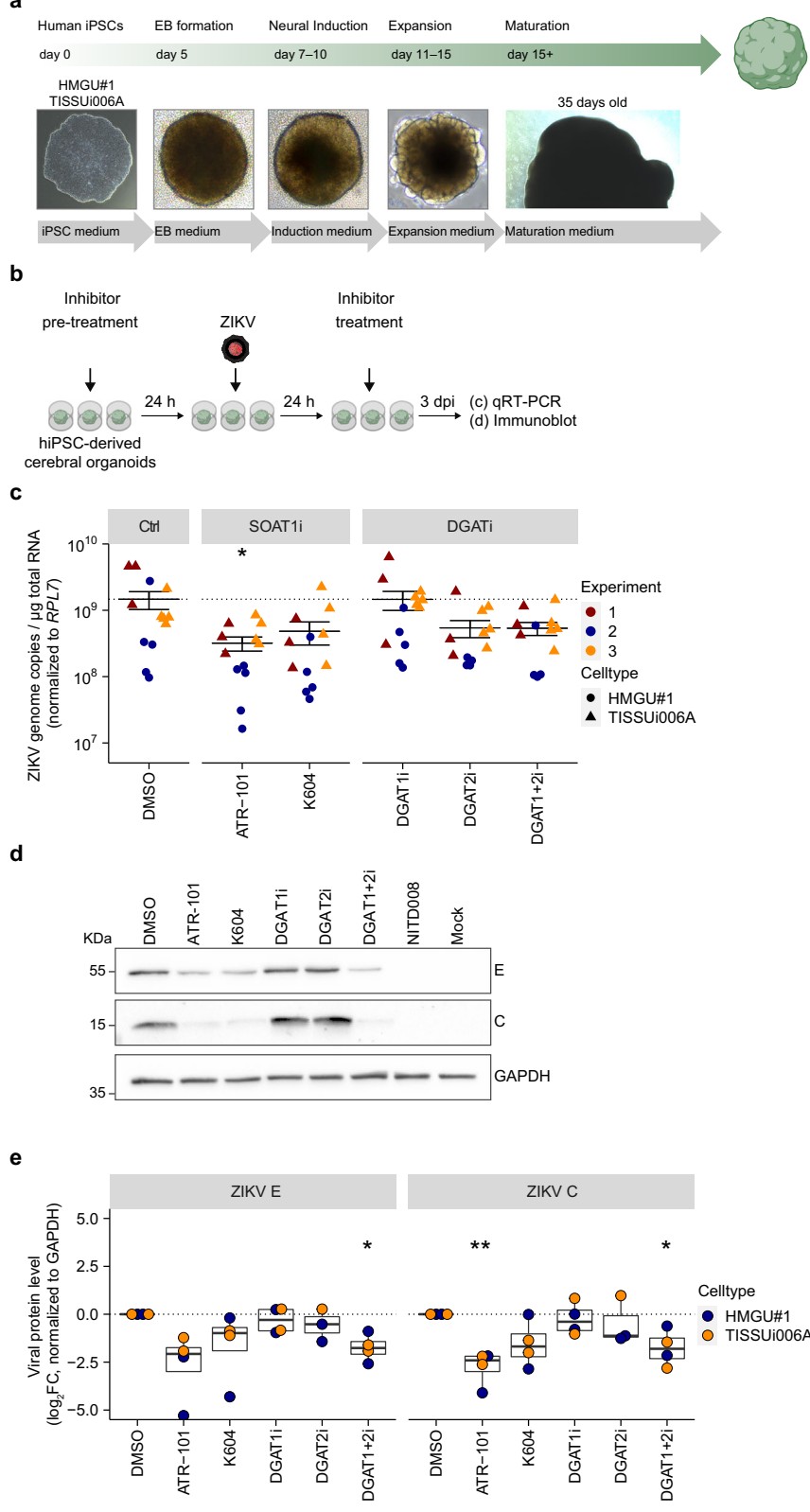

nucleocapsid protein core[27]. During HCV replication, DGAT1 activity is essential to localize HCV core to LDs and subsequently for efficient particle assembly. Although DGAT1 activity seems to be dispensable for ZIKV replication in our cell culture systems, the direct interaction seems to be important to facilitate efficient replication but the exact mechanism remains to be elucidated.

Interestingly, inhibition of both DGAT enzymes at once strongly blocked ZIKV replication in cerebral organoids, which represent a complex model with various neural cell types, including astrocytes, neurons, and neural progenitor cells. Neurons and glia cells display a strong crosstalk in order to maintain function, including lipid transfer between both cell types[44]. Upon oxidative stress, neurons transfer excess fatty acids to

astrocytes via lipoproteins[45,46], and LD formation in astrocytes, but not in neurons, increases under hypoxic conditions[45–47]. Given that ZIKV infection can generate oxidative stress[48–50], complete inhibition of TG synthesis might result in a cellular lipid imbalance that is unfavorable for ZIKV replication.

In contrast to DGAT, inhibition of SOAT1, especially with the more potent ATR-101 inhibitor, reduced ZIKV infection in all our cell culture models, indicating that SOAT1 activity is critical to ZIKV. In line, pharmacological inhibition of cholesterol esterification also decreased HBV[24], and HCV replication[25].

Given that ZIKV replication did not recover in SOAT1i-treated cells after cholesteryl oleate treatment, our data point to a cholesterol-dependent effect rather than to cholesteryl esters being involved. A common feature in flaviviral replication organelle formation is remodeling of the ER membrane[5]. Free cholesterol was found to accumulate at DENV and WNV as well as HCV replication sites[7,18,19,51,52] and statin treatment reduced ZIKV, WNV, DENV, and HCV infection[19–21,53,54]. On the other hand, excess cholesterol decreased WNV, DENV, and JEV infection, implicating that cholesterol homeostasis is essential for orthoflavivirus replication[19,22]. Blocking cholesterol esterification leads to an imbalance in cellular cholesterol levels or distribution that may be detrimental for ZIKV infection. As we did not detect any changes in ZIKV dsRNA foci in SOAT1i-treated Hmc3 cells, viral RNA replication seems to be not affected. In line, SOATi treatment did not change HCV dsRNA foci, but decreased the production of infectious particles[25]. Reduced E protein and ZIKV RNA level in supernatants of SOAT1i-treated cells indicate that lack of SOAT1 activity also impairs ZIKV particle production. In contrast to HCV[25] the specific infectivity of secreted ZIKV particles was reduced, indicating defective particle production or egress; however, the exact mechanism needs to be investigated. With regard to SOAT2, SARS-CoV-2 replication was suppressed by treatment with Avasimibe, a SOAT1/2 inhibitor, and additional knockdown experiments highlighted a predominant role for SOAT2 compared to SOAT1 in SARS-CoV-2 replication in Vero and Calu-3 cells[26]. Due to the cytotoxic effect of SOAT2 depletion or inhibition, we were not able to address a potential role of SOAT2 in ZIKV replication in Huh7 cells.

Suppression of ZIKV replication by SOAT1i was much stronger in Hmc3 and 1321N1 cells compared to Huh7 cells. While liver cells express SOAT2 in addition to SOAT1 and SOAT2 is the main cholesterol-esterifying enzyme in the liver, SOAT1 is the only isoform expressed in brain cells[29,55–57]. Thus, SOAT2 might partially compensate the lack of SOAT1 activity in Huh7 cells, explaining the lesser effects on ZIKV replication in hepatoma cells compared to Hmc3 and 1321N1 cells. Other additional factors might include further differences in lipid metabolism or stability of the inhibitors. Previous reports have shown increased levels of free cholesterol in cells with inhibited cholesterol esterification[26,32]. Although measurements of free cholesterol in cell lysates revealed a marginal increase in ATR-101-treated cells only, we observed a strong intracellular accumulation of free cholesterol in SOAT1i-treated Huh7 and Hmc3 cells regardless of which SOAT1i we used by filipin III staining. Addition of cholesterol during DENV and JEV infection blocked viral entry and intracellular replication steps[22] and WNV infection[19]. In line, treatment with cholesterol during entry reduced ZIKV CPE in Huh7 and Hmc3 cells, but small plaques were still formed. Adding cholesterol post-infection completely blocked ZIKV replication in Hmc3 cells but even slightly elevated it in Huh7 cells. In contrast to Hmc3, Huh7 cells might rapidly esterify cholesterol thereby alleviating its effect in ZIKV infection. This may also explain why we observed the most profound effect of SOAT1 inhibition in Hmc3 cells, as they likely accumulate plenty of free cholesterol as a consequence of its blocked esterification.

Recently, K604-treatment was shown to increase cholesterol at mitochondria-associated membranes (MAMs), concurrent with enrichment of SOAT1 in these subcellular structures, and increased mitochondria-ER contacts[58]. Since ZIKV infection induces ER-remodeling[59], alterations in organelle contacts might counteract virus-induced membrane adjustments necessary for virus replication.

In conjunction with their role in ZIKV replication, we identified DGAT1 and SOAT1 as host factors for WNV, TBEV, and DENV infection with the strongest effect observed for DENV and the least effect on TBEV. Taken together, we identified neutral lipid–synthesizing enzymes as important host dependency factors for orthoflavivirus replication. Further, our findings highlight SOAT1 as druggable host factor for ZIKV infection in various cell types and point towards a potential pan-orthoflavivirus target.

## Methods
### Reagents
The following antibodies and reagents used in this study were obtained commercially: Flavivirus group antigen antibody (D1-4G2-4-15 (4G2), NBP2-52666, Novus Biologicals), Flavivirus NS1 antibody ((D/2/D6/B7), ab214337, abcam), Zika virus capsid protein antibody (GTX133317), SOAT1 antibody (GTX32890) (all GeneTex), GAPDH antibody (clone G-9, sc-365062), DGAT1 antibody (clone H-255, sc-32861) (all Santa Cruz Biotechnology), J2 dsRNA antibody (Scicons/Jena Biosciences RNT-SCI-10010200), PLIN2 antibody (610102, Progen), PLIN3 antibody (HPA006427, Sigma), beta-actin antibody (clone AC-74, A2228-200UL, Sigma), FLAG antibody (F7425, Sigma), Strep-tag II antibody (ab184224, abcam), HRP-labelled secondary antibodies (Jackson ImmunoResearch), Anti-Rabbit IgG HRP conjugated (Rabbit TrueBlot) (clone eB182, 18-8816-33, Rockland Immunochemicals), Alexa647-conjugated secondary antibody (donkey, IgG (H + L), A-31571), BODIPY493/503 (D-3922), BODIPY 655/676 (B-3932), Hoechst 33342 (H1399), Cholesteryl BODIPY FL C12 (C3927MP) (all Invitrogen), filipin III (F4767), K-604 (SML1837), ATR-101 (SML2802), DGAT1 inhibitor (PF-04620110), DGAT2 inhibitor (PF-06424439), NITD 008 (SML2409), cholesteryl oleate (C9253), cholesterol-water soluble (~ 40 mg of cholesterol per gram; balanced with methyl-β-cyclodextrin) (C4951), oleic acid-BSA complex (O3008) (all Sigma). The Rho-kinase Y-27632 inhibitor (ROCK inhibitor) was purchased from Tocris. Geltrex ESC / hiPSC-qualified, LDEV-free, growth factor reduced basement membrane matrix was purchased from Gibco and Matrigel LDEV-free basement membrane matrix was purchased from Corning. LD540 was a gift from Christoph Thiele. Enzymes were purchased from Thermo Fisher Scientific, restriction enzymes for molecular cloning from NEB. Primer and oligonucleotides were ordered from Sigma. If not noted otherwise, chemicals were purchased from Applichem or Sigma/ Merck and cell culture reagents were used from Gibco/Thermo Fisher Scientific. Avicel was purchased from FMC Biopolymer Germany GmbH.

### Cell lines and generation of cerebral organoids from hiPSCs
HEK293T and VeroE6 cells were obtained from the American Type Culture Collection, and Huh7.5 cells from Apath, LCC. The 1321N1 cell line (86030402) and the SH-SY5Y cell line (94030304) were purchased from Merck. Huh7 cells were provided by Ralf Bartenschlager and BHK-21 cells by César Muñoz-Fontela. Hmc3 cells were provided by Alexander Slowik. HMGU#1 and TISSUi006A human induced pluripotent stem cells (hiPSCs) were obtained from the Pluripotent Stem Cell core facility at the Helmholtz Institute, Munich and from the organoid platform at the Max-Delbrück Institute, Berlin, respectively. Cell lines were grown in high-glucose DMEM supplemented with 10% FBS and 1% GlutaMax under standard cell culture conditions. Cell viability assays were performed using the CellTiter 96 AQueous One Solution Reagent (Promega). hiPSCs were cultivated on feeder-free conditions in previously prepared 6-well plates coated with Geltrex (Life Technologies) at 1:100 dilution in DMEM-F12 overnight at 4 °C. mTeSR plus medium (Stem Cell Technologies) was used for routine cultivation of hiPSCs. Cells were daily checked for areas of spontaneous differentiation as well as routinely checked for mycoplasma contamination. Cells were passaged at confluency of 70–80% using ReLeSR (Stem Cell Technologies).

Cerebral organoids were generated using the StemDiff Cerebral organoid kit (Stem Cell Technologies) that is based on a previously published protocol[60,61]. Briefly, hiPSCs were cultured until cultures reached a confluency of 70%. On day 0, hiPSCs were dissociated into single cells using

the gentle dissociation reagent (Stem Cell Technologies). The single cell suspension was resuspended in embryoid body (EB) formation medium supplemented with 50 μM of Rho-associated protein kinase (ROCK) inhibitor and seeded into ultra-low attachment 96-well plates at a density of 9000 cells per well. On day 2, 100 μl of EB formation medium supplemented as described above were added to the wells. On day 5, EBs were transferred to fresh ultra-low attachment plates with 100 μl of neural induction medium. On day 6 to 8 EBs that showed clear signs of neural induction (formation of translucent and radially organized neuroepithelia) were embedded in hESC/iPSC-qualified Matrigel droplets and cultivated in expansion medium for 3–4 days until prominent neuroepithelia expansion was formed and visible. From day 15 on, organoids were cultivated in 60 mm ultra-low attachment petri dishes with maturation medium on an orbital shaker at 65 rpm. Approximately on day 35, organoids were used for further analysis and infection experiments.

## Plasmids

Two-plasmid infectious clone systems used for ZIKV PRVABC59 (plasmids pJW231 and pJW232[62]) and WNV-NY99 (plasmids pWN-AB and pWN-CG[63]), and plasmids encoding full length TBEV Neudoerfl (plasmid pTNd/c[64]) and DENV-2 16681[65] were described previously.

The following lentiviral vectors were described previously: pSicoR-MS1 shDGAT1 and pSicoR-MS1 shDGAT2[27], and pSicoR-MS1 shNT[66] (target sequences: shDGAT1 GGAACATCCCTGTGCACA, shDGAT2 GCGAAAGCCACTTCTCATA, shNT GCGCGATAGCGCTAATAATT). shSOAT1 was cloned into pSicoR-MS1 using HpaI and XhoI restriction sites and shSOAT1 fw TTAATCACCTCCACTCATATTATTCAA GAGATAATATGAGTGGAGGTGATTATTTTTTC and shSOAT1 rev TCGAGAAAAAATAATCACCTCCACTCATATTATCTCTTGAATAA TATGAGTGGAGGTGATTAA oligonucleotides (target sequence: shSOAT1 TAATCACCTCCACTCATATTA) as described[27,67]. The following expression constructs were previously described: FLAG-tagged murine DGAT1[68] and ZIKV Uganda 1947 MR766 C (-anchor)-2x Strep II expression plasmid[69]. To clone a ZIKV PRVABC59 C-2x Strep II expression plasmid, the ZIKV PRVABC59 C ( + anchor) was amplified from pJW231 using EcoRI_ZIKV PRV C fw GTGGAATTCATGAAAAACCCAAAA and ZIKV PRV C +anchor_XhoI rev CACCTCGAGTGCCATAGC TGTGGT primers. Using EcoRI and XhoI restriction sites, the ZIKV Uganda 1947 MR766 C ORF in ZIKV Uganda 1947 MR766 C (-anchor) -2x Strep II expression plasmid was replaced by the ZIKV PRVABC59 C ORF. For the ZIKV EGFP reporter construct previously described[31] we exchanged the DENV capsid protease cleavage site (RRRRSAGM) by the corresponding capsid cleavage site (KKRRGADT) of ZIKV strain HPF2013 (GenBank # KJ776791.2). We ordered a GBlock (IDT) covering the coding sequence for EGFP, the nuclear localization signal of the SV40 large T-antigen, the cleavage site for the ZIKV protease and the transmembrane domain of the ER resident protein Sec61β. We cloned this fragment by Gibson assembly into the pWPI-Neo vector using the AscI and SpeI sites to linearize the vector. The nucleotide sequence of the insert is available upon request.

## Immunoblot analysis

Cells were lysed in RIPA lysis buffer (150 mM NaCl, 50 mM Tris/HCl pH 7.4, 1% Nonidet-P40, 0.5% sodium deoxycholate, 1 mM EDTA, 0.1% SDS, supplemented with 1x protease inhibitor cocktail (Sigma) and 1 mM phenylmethylsulfonyl fluoride (PMSF) for 60 min on ice. Lysates were clarified by centrifugation and subjected to SDS-PAGE followed by transfer to a nitrocellulose membrane (GE Healthcare). Samples for detection of SOAT1 and DGAT1 were not heated before SDS-PAGE to prevent aggregation of the respective proteins. If membranes were to be probed with antibodies for orthoflavivirus group antigen or orthoflavivirus NS1, Laemmli buffer without β-mercaptoethanol was used. Supernatants of infected cells were mixed with Laemmli buffer, heated, and equal volumes were subjected to SDS-PAGE and immunoblot analysis. Membranes were probed with the respective antibodies and bands were detected by chemiluminescence using

Immobilon Classico or Immobilon Forte (Merck), Lumi-Light substrate (Roche), or SuperSignal West Femto (Thermo Fisher) and the Image Lab system (BioRad). Signal intensities of specific bands were analyzed using the Image Lab quantification function.

## Co-Immunoprecipitation

Cells were lysed in 1% NP-40 lysis buffer (50 mM Tris, pH 7.4, 150 mM NaCl, 1% Nonidet-P40) supplemented with 1x protease inhibitor cocktail and 1 mM PMSF for 30–60 min on ice. Prior to clarification, lysates were passed 10 x through a 23 G needle. Pre-clearing was performed for 30 min at 4 °C, rotating, using rProtein G agarose (Invitrogen). FLAG-specific immunoprecipitation was performed using anti-FLAG M2 affinity gel (Sigma) for 2.5 h at 4 °C, rotating. Subsequently, beads were washed 4 times in cold 1% NP40 lysis buffer before proteins were eluted in 3x Laemmli buffer and analyzed by immunoblotting.

## Immunofluorescence and microscopy

For immunofluorescence analysis cells grown on cover slips were fixed with 4% paraformaldehyde, permeabilized in 0.1% Triton-X-100/PBS for 5 min and incubated in blocking solution (5% BSA, 1% fish skin gelatin, 50 mM Tris in PBS). Primary antibody staining in blocking solution was performed overnight, followed by incubation with Alexa Fluor-coupled secondary antibodies. LDs were visualized using BODIPY493/503 or BODIPY665/676 and nuclei were stained with Hoechst. Coverslips were embedded in mowiol mounting media and analyzed by confocal microscopy on a Leica Stellaris 8 confocal laser scanning microscope. For dsRNA analysis, Hmc3 cells stably expressing a cell-based ZIKV EGFP reporter were used. dsRNA foci were quantified using the Particle Analyzer function of Fiji[70]. For visualization of free cholesterol, fixed cells were washed with PBS, quenched with 1x TBS (50 mM Tris, 150 mM NaCl, pH 7.5) for 5 min at RT, and stained with filipin III (0.1 mg/ml filipin III in 5% BSA/TBS). After additional washing steps, LDs were visualized with BODIPY493/503 or LD540 and samples were mounted in mowiol mounting media. Microscopy was performed on a Leica Thunder Cell Imager. Signal intensities of individual cells were quantified with Fiji. Here, single cells were selected by drawing a region of interest (ROI), and the cellular total corrected fluorescence (CTCF) was calculated from the integrated density of the filipin III channel and the cell area.

## Electron microscopy

For electron microscopy, cells were grown on sapphire discs (Engineering Office M. Wohlwend GmbH, Sennwald, Switzerland). The cells were fixed for 1 h in 0.1 M Na-cacodylate buffer pH 7.2 containing EM-grade 2.5% glutaraldehyde and 2% formaldehyde, followed by washing and incubation in 1% osmium tetroxide and 1.5% ferricyanide for 1 h in the same buffer. The cells were incubated in 2% uranyl acetate in water over night, dehydrated through an ethanol series into acetone, and subsequently infiltrated with resin (EMbed-812 Kit with BDMA), and polymerized at 60 °C for 48 h. The sapphire disc was removed to expose the cells at the blockface. Sections with a nominal thickness of 70 nm were cut using a Leica EM UC6 ultra-microtome and an ultra 45° diamond knife (Diatome), and mounted on formvar-coated single slot nickel grids. Sections were subsequently stained with 2% uranyl acetate followed by 3% lead citrate. Samples were imaged at an acceleration voltage of 120 kV in a JEOL JEM-1400 TEM equipped with a TemCam-F416 camera (TVIPS, Gauting, Germany; unless otherwise mentioned, the materials are from Science Services, Munich, Germany).

## RNA isolation and quantitative RT-PCR

Total RNA from cells and cerebral organoids was isolated using TriReagent (Sigma) and viral RNA from supernatants was isolated using the NucleoSpin RNA Virus Kit (Macherey-Nagel). Organoid samples were vortexed with Lysing Matrix H (MPbio) in TriReagent until the tissue was completely dissolved and glycogen (RNA grade, Thermo Fisher) was added as carrier. Residual DNA was removed by rDNAseI-treatment (DNA-free Kit, Invitrogen), and RNA was reversely transcribed using Superscript III

reverse transcriptase (Invitrogen), RNAseOut (Invitrogen), and random hexamer primers (Qiagen). qRT-PCR analysis was performed using the Luna Universal qPCR Master Mix (NEB) on a StepOne Plus Real time PCR system or a Quant Studio 3 (Applied Biosystems). The following qRT-PCR primers were used: SOAT1 fw GAAGTTGGCAGTCACTTTGATGA; SOAT1 rev GAGCGCACCCACCATTATCTA; DGAT1 fw TATTG CGGCCAATGTCTTTGC; DGAT1 rev CACTGGAGTGATAGACTCAA CCA; DGAT2 fw ATTGCTGGCTCATCGCTGT; DGAT2 rev GGGAAA GTAGTCTCGAAAGTAGC; 18S rRNA fw GTAACCCGTTGAACCC-CATT; 18S rRNA rev CCATCCAATCGGTAGTAGCG; GAPDH fw AAGGTGAAGGTCGGAGTCAAC; GAPDH rev GGGGTCATT-GATGGCAACAATA; RPL7 fw CCAAATTGGCGTTTGTCAG; RPL7 rev GCATGTTCGAAGCCTTGTTG; (all Harvard primer bank); ZIKV fw TTCGGAATATGGAGGCTGAG; ZIKV rev TCGTTTGAGCCTAT CCCATC[71]; RPL7 fw CCAAATTGGCGTTTGTCAG; RPL7 rev GCATGT TCGAAGCCTTGTTG; SOX2 fw GCCGAGTGGAAACTTTTGTCG; SOX2 rev GGCAGCGTGTACTTATCCTTCT; PAX6 fw GTCCATCT TTGCTTGGGAAA; PAX 6 rev TAGCCAGGTTGCGAAGAACT; S100B fw TGGCCCTCATCGACGTTTTC; S100B rev ATGTTCAAAGAAC TCGTGGCA; TUBB3 fw GGCCAAGGGTCACTACACG; TUBB3 rev GCAGTCGCAGTTTTCACACTC; MAP2 fw CGAAGCGCCAATG-GATTCC; MAP2 rev TGAACTATCCTTGCAGACACCT; NEUN fw CCCATCCCGACTTACGGAG; NEUN rev GCTGAGCGTATCTG-TAGGCT all[72].

## Lentivirus production
Lentivirus stocks were produced in HEK293T cells as described[66,73]. Briefly, cells were co-transfected with the pSicoR-MS1 shRNA plasmids, a lentiviral packaging vector (pCMV delta R8.91) and a vector expressing the VSV-G protein (pMD.G). 3 days post transfection lentiviral pseudoparticles were harvested and concentrated by ultracentrifugation at 27,000 rpm, 4 °C in a SW-32 rotor (Beckman-Coulter). Lentiviral stocks were titrated on Huh7 cells and transduction was performed in medium supplemented with 4 μg/ml polybrene.

## In vitro transcription of orthoflavivirus RNA and virus stock production
To prepare viral stocks, DENV-2 16681 and TBEV Neudoerfl-full length encoding plasmids were linearized using XbaI or NheI, and purified by phenol-chloroform extraction. Linear full-length DNA of ZIKV PRVABC59 and WNV-NY99 was generated from the two-plasmid system by overlap extension PCR using the following primers: ZIKV pJW231 BamHI fw TAGGATCCTAATACGACTCACTATAG; ZIKV Conserved 3499 rev[74], GCCTTATCTCCATTCCATACCA; ZIKV pJW232 ApaLI fw AGGGGAGTGCACAATGCCCCCA; ZIKV pJW232 EcoRI rev CTAGAA TTCGCCCTTGCTCCG; WNV AB fw CATCTCAGCTCTTGCCGGCT-GATGTCTATGGCAC; WNV AB rev ACGCGTAAATTTAATACG ACT; WNV CG fw TCTAGAGATCCTGTGTTCTC; WNV CG rev GCCGGCAAGAGCTGAGATGT. In vitro transcription of DENV-2 RNA was performed with the MEGAscript SP6 Transcription Kit and a (m7G (5') ppp (5')G) Cap analog (Invitrogen), for all other in vitro transcriptions the HiScribe T7 ARCA mRNA Kit (NEB) was used. Purified RNA was transfected into VeroE6 cells (DENV-2, WNV-NY99, ZIKV PRVABC59) or BHK-21 cells (TBEV Neudoerfl) by electroporation. Here, $4 \times 10^6$ cells were washed in OptiMEM and the cell pellet was resuspended in 400 μl cytomix buffer (120 mM KCl, 5 mM MgCl$_2$, 0.15 mM CaCl$_2$, 2 mM EGTA, 25 mM HEPES, 10 mM potassium phosphate buffer, pH 7.6) supplemented with 1.9 mM ATP and 4.7 mM glutathione. Cells were added to 5–10 μg in vitro transcribed orthoflavivirus RNA and pulsed at 260 V and 950 μF using the Gene Pulser II (Biorad)[66]. Cells were cultured in medium supplemented with 10–15 mM HEPES until cytopathic effects were evident. Supernatants were harvested, filtered, and stored at –80 °C. To produce p1 viral stocks, supernatant from electroporated cells was used to infect naïve cells.
For organoid infection ZIKV stocks were produced in serum-free Opti-MEM in order to avoid FBS growth factor interference in organoid

expansion and growth. For this, ZIKV PRVABC59 p1 produced as described above was used to inoculate semi-confluent naïve VeroE6 cell for 1 h at 37 °C. Thereafter, the inoculum was removed, cells were washed once with 1x PBS and cultured in serum-free Opti-MEM for 5–6 days until cytopathic effects were visible. Supernatants were harvested, clarified and stored at –80 °C until further use. Viral stocks for organoid infection were titrated by plaque assay on Vero cells.

## Orthoflavivirus infection
For infection, cell culture media was removed, and cells were infected with virus stocks diluted in media supplemented with 10 mM HEPES for 1 h at 37 °C. The virus inoculum was removed and replaced by media supplemented with 10 mM HEPES. For analysis of ZIKV replication in inhibitor-treated cells, pre-treatment was performed for 24 h before infection. After the inoculum was removed, cells were cultured in media containing the inhibitors for 48 h before samples were taken. For analysis of ZIKV infection in inhibitor-treated cerebral organoids 30–35 days old organoids were pre-treated with inhibitors for 24 h. Organoids were infected with ZIKV PRVABC59 at an MOI of $10^5$ TCID$_{50}$ per organoid in inhibitor-supplemented cerebral organoid maturation medium The next day, the inoculum was removed, organoids were washed once with 1x PBS, and fresh cerebral organoid maturation medium with the respective inhibitors was added to the dishes and cultivated as described above.

## TCID$_{50}$ assays
To compare virus production, the 50% tissue culture infectious dose (TCID$_{50}$) was analyzed. Briefly, BHK21 cells were seeded in 96 well plates, infected with 10-fold serial dilutions of virus-containing supernatants in triplicates and incubated for 6 days. After PFA-fixation, cells were stained with crystal violet and CPE-positive wells were counted. The TCID$_{50}$ / ml was calculated using the Reed & Munch calculator[75]. Viral stock titers were determined on Huh7 cells (DENV-2, WNV-NY99, ZIKV PRVABC59) or BHK-21 cells (TBEV Neudoerfl) at 5 dpi.

## Plaque size reduction assays
shRNA-transduced Huh7.5 cells were seeded in 6 well plates with $4–4.5 \times 10^5$ cells / well. The following day, cells were infected with 10-fold serial dilutions of ZIKV PRVABC59 stocks. After 1 h, the inoculum was replaced by Avicel overlay media (1.25% Avicel in 1x MEM + 10% FCS + 1% Pen/Strep). Cells were fixed with 4% PFA at 4 dpi and stained with 0.1% crystal violet in 10% ethanol to visualize plaque formation.

## CPE reduction assays
Huh7 knockdown cells were seeded in 12 well plates with $8 \times 10^4$ cells / well and infected with ZIKV PRVABC59 the following day. The inoculum was removed after 1 h, and cells were incubated for 96 h. For inhibitor treatments, $4 \times 10^4$ Huh7 or Hmc3 cells were seeded, treated the following day, and infected with ZIKV PRVABC59 24 h post initial treatment. The inoculum was replaced by inhibitor-supplemented media after 1 h, and cells were fixed at 3–4 dpi. Cells were stained with 0.1% crystal violet in 10% ethanol to visualize CPE. Crystal violet staining was quantified with Fiji.

## Entry Assay
Viral entry efficacy was determined as described[76] with minor modifications: Hmc3 cells were pre-treated for 24 h with 0.2% DMSO or SOAT1i (20 μM K604 or 9.72 μM ATR-101). Prior to infection, the media was changed to cold DMEM supplemented with 3% FCS and the cultures were pre-cooled for 10 min at 4 °C prior to infection with ZIKV (MOI 5). Inoculation was performed for 1 h at 4 °C before cells were washed with cold DMEM containing 10% FCS. Fresh media supplemented with DMSO or SOAT1i was added and cells were shifted to 37 °C for 4 h. After washing 1x with DMEM and 1x with PBS, incubation with pre-warmed trypsin for 2 min at RT was performed to remove remaining attached virions. Subsequently, cells were washed again with DMEM and PBS and lysed in

TriReagent (Sigma-Aldrich). RNA Isolation and RT-qPCR were performed as described above.

## Measurement of free cholesterol

To determine free cholesterol in SOAT1-depleted or SOAT1i-treated cells, $4 \times 10^4$ knockdown or naïve cells were seeded in 24 wells. The following day, naïve cells were treated with K604, ATR-101 or DMSO and free cholesterol was analyzed using the Cholesterol/Cholesterol Ester-Glo Assay (Promega) at 48 h post-treatment, 5 days post-transduction, respectively.

## Statistics and reproducibility

Data was analyzed using R and R studio (R Core Team, 2015; R Studio Team 2020). Bar graphs are shown as mean ± standard error of the mean (SEM) and box-and-whisker plots display center line: median, box limits: upper and lower quartiles, whiskers: 1.5 x interquartile range, points: outliers. For statistical analysis we used an unpaired two-tailed *t*-test with unequal variance (Welch's *t*-test) or a one-sample *t*-test in case of normalized data. The number of independent experiments conducted and the statistical method applied to each data set are described in the individual figure legends. If not noted otherwise, sample size (n) represents individual experiments.

## Reporting summary

Further information on research design is available in the Nature Portfolio Reporting Summary linked to this article.

## Data availability

Primary data generated and analyzed in this study are available upon request from the corresponding author. The source data behind the graphs in all main and Supplementary Figs. can be found in Supplementary Data 1. The uncropped immunoblots can be found in the Supplementary Information document (Supplementary Fig. 8).

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

## Acknowledgements

We thank Melanie Ott and Sakshi Tomar (Gladstone Institutes, CA) and Greg Ebel (Colorado State University) for the ZIKV PRVABC59 plasmids, Claire Huang (CDC, GA) for the WNV-NY99 two plasmid system, Karin Stiasny (Medical University of Vienna, Austria) for the TBEV Neudoerfl pTNd/c plasmid, Nevan J. Krogan (Gladstone Institutes, CA) for the ZIKV Uganda 1947 MR766 C expression plasmid, Robert V Farese Jr. (Sloan Kettering Institute, NY) for the FLAG-tagged murine DGAT1 expression plasmid, Ralf Bartenschlager (University of Heidelberg, Germany) for DENV-2 16681 construct and Huh7 cells, Charles M. Rice (Rockefeller University, NY) for Huh7.5 cells, César Muñoz-Fontela (BNITM, Hamburg, Germany) for

BHK-21 cells, Alexander Slowik (RTWH Aachen, Germany) for Hmc3 cells, Ejona Rusha (Helmholtz Institute, Munich, Germany) for HMGU#1 hiPSCs, Agnieszka Rybak-Wolf (Max-Delbrück Center, Berlin, Germany) for TIS-SUi006A hiPSCs, and Matt Spindler (Gladstone Institutes, CA) for pSicoR-MS1. This study was supported by funds from the Deutsche Forschungsgemeinschaft [DFG project IDs 416701689, 517270163, 517270083, 197785619 (CRC 2021 project B10) to E.H.] and the LOEWE Center DRUID (Novel Drug Targets against Poverty-related and Neglected Tropical Infectious Diseases, E.H.).

## Author contributions

Conceptualization, A.S., E.H. Methodology, A.S., E.H. Investigation, A.S., R.B., V.R., M.S. Writing – Original Draft, A.S., E.H. Writing – Review & Editing, A.S., R.B., V.R., J.H., L.S., M.S., G.V., E.H. Funding Acquisition, E.H. Resources, R.B., J.H., L.S., V.R., G.V. Supervision, E.H.

## Funding

## Competing interests

The authors declare no competing interests.

## Additional information

**Peer review information** *Communications Biology* thanks the anonymous reviewers for their contribution to the peer review of this work. Primary handling editors: Christina Karlsson Rosenthal and Manuel Breuer. A peer review file is available. This manuscript has been previously reviewed at another Nature Portfolio journal. This document only contains reviewer comments and rebuttal letters for versions considered at Communications Biology.

