## [Peer Review File · Communications Biology]

REVIEWERS' COMMENTS:

Reviewer #1 (Remarks to the Author):

This is a much improved manuscript from the authors, and I feel that the new data greatly strengthens the manuscript and its claims. The authors have answered all my queries and taken on board suggestions from both reviewers. I only have 2 comments on the newer data:

1. Figure 6d: As there is no specific labelling of structures in your TEM, I feel that a panel with uninfected cells should be shown to confirm for your readers (especially the non-experts in flaviviruses), the absence of VI and VP (Particularly virions).

Minor comment:

Figure legend for figure 6: line 919 appears to be missing a word 'ZIKV RNA copies in supernatants of were determined...'

Response to the reviewers' concerns

We thank the reviewers for their final assessment of our manuscript. We now addressed the remaining concerns of reviewer #1.

Reviewer #1 (Remarks to the Author):

This is a much improved manuscript from the authors, and I feel that the new data greatly strengthens the manuscript and its claims. The authors have answered all my queries and taken on board suggestions from both reviewers. I only have 2 comments on the newer data:

1. Figure 6d: As there is no specific labelling of structures in your TEM, I feel that a panel with uninfected cells should be shown to confirm for your readers (especially the non-experts in flaviviruses), the absence of VI and VP (Particularly virions).

We thank the reviewer for his/her comment. We now added transmission electron microscopy images of a ZIKV-infected and an uninfected Hmc3 cell to distinguish between virus-induced structures and cellular compartments. We also added labels to the images for clarification (**New Supplementary Figure 3**).

Minor comment:

Figure legend for figure 6: line 919 appears to be missing a word 'ZIKV RNA copies in supernatants of were determined...'

We thank the reviewer for his/her observation. We now edited the respective line "ZIKV RNA copies in supernatants of SOAT1i-treated cells were determined..."